# FedFree: Breaking Knowledge-sharing Barriers through Layer-wise Alignment in Heterogeneous Federated Learning

**Haizhou Du**[*][*]   **Yiran Xiang**[*]   **Yiwen Cai**[*]

**Xiufeng Liu**[*]   **Zonghan Wu**[†]   **Huan Huo**[‡]   **Guodong Long**[‡]

[*]*Shanghai University of Electric Power, Shanghai, China*
[*]*Technical University of Denmark, Kongens Lyngby, Denmark*
[†]*East China Normal University, Shanghai, China*
[‡]*University of Technology Sydney, Sydney, Australia*

## Abstract

Heterogeneous Federated Learning (HtFL) enables collaborative learning across clients with diverse model architectures and non-IID data distributions, which are prevalent in real-world edge computing applications. Existing HtFL approaches typically employ proxy datasets to facilitate knowledge sharing or implement coarse-grained model-level knowledge transfer. However, such approaches not only elevate risks of user privacy leakage but also lead to the loss of fine-grained model-specific knowledge, ultimately creating barriers to effective knowledge sharing. To address these challenges, we propose FedFree, a novel proxy-data-free and model-free HtFL framework featuring two key innovations. First, Fed-Free introduces a reverse layer-wise knowledge transfer mechanism that aggregates heterogeneous client models into a global model solely using Gaussian-based pseudo-data, eliminating reliance on proxy datasets. Second, it leverages Knowledge Gain Entropy (KGE) to guide targeted layer-wise knowledge alignment, ensuring that each client receives the most relevant global updates tailored to its specific architecture. We provide rigorous theoretical convergence guarantees for FedFree and conduct extensive experiments on CIFAR-10 and CIFAR-100. Results demonstrate that FedFree achieves substantial performance gains, with relative accuracy improving up to 46.3% over state-of-the-art baselines. The framework consistently excels under highly heterogeneous model/data distributions and in large-scale settings.

## 1  Introduction

Heterogeneous Federated Learning (HtFL) represents a significant advancement in federated learning (FL), addressing the prevalent challenges of statistical and model heterogeneity encountered in real-world edge computing scenarios [1]. In these settings, participating clients often possess devices that have limited resources with varying computational capabilities, necessitating the use of diverse local model architectures tailored to specific needs [2, 3]. This inherent heterogeneity, however, introduces significant **knowledge-sharing barriers**, as knowledge encoded within disparate model structures is difficult to effectively aggregate and disseminate across the federation.

Existing HtFL methodologies often attempt to bridge this knowledge-sharing gap using techniques like knowledge distillation [4, 5, 6, 7, 8, 9, 10, 11, 12], representation and prototype shar-

---

[*]Email: haizhou.du@shiep.edu.cn

ing [13, 14, 15, 16, 17] and architectural adaptation and matching [18, 19, 20, 21, 22, 23]. These methods have overcome the bottlenecks imposed by client resource limitations, enabling more efficient model collaboration and training across devices in heterogeneous federated learning. However, knowledge distillation transfers knowledge across models via proxy data but risks privacy leakage. Representation/prototype sharing exchanges intermediate features or class prototypes lacks fine-grained layer-wise adaptation. Architectural adaptation directly aligns heterogeneous structures through matching or compression, yet faces suboptimal knowledge retention problems.

These limitations reveal two fundamental and persistent challenges in HtFL. The key challenge is how to share personalized knowledge without the proxy dataset, causes local-to-global knowledge-sharing barrier. Traditional knowledge transfer-based methods fail to obtain learned knowledge without the proxy dataset. Eventually, it leads to the failure of collaborative training among cross-clients. The other important challenge is how to decide that the corresponding knowledge is distributed to well-suited clients, causes global-to-local knowledge-sharing barrier. Conventional methods always directly distribute the transferred global model to heterogeneous local models using simple strategies.

To overcome these challenges, we propose **FedFree** (**Fed**erated Proxy-Data-**free** and Model-**free**), a novel HtFL framework that operates entirely without proxy data. FedFree introduces two core innovations: First, it employs a **reverse layer-wise knowledge transfer** mechanism. Rather than the server dictating knowledge to clients, FedFree identifies critical layers (defined in 3.3) from local models as the source of knowledge, transferring layer-wise local knowledge to the global model layers. This local-to-global flow is facilitated by **Gaussian-based pseudo-data** generated on the server, aiming to elicit functional responses for local and global layer matching, thereby circumventing the need for proxy data. Second, for the global-to-local knowledge distribution, FedFree introduces **Knowledge Gain Entropy (KGE)**, a novel metric designed to quantify the knowledge density difference between corresponding global and local layers. By comparing layers of the same type (*e.g.*, convolutional-to-convolutional) using KGE, the server identifies and transmits global layers offering the high-density knowledge to each client, enabling targeted and efficient knowledge alignment.

The **main contributions** of this paper are summarized as follows:

- We propose FedFree, a novel HtFL framework that addresses the knowledge-sharing challenges between heterogeneous clients and the global model, by enabling proxy-data-free and model-free solutions for fine granular knowledge alignment.

- We introduce a *reverse layer-wise knowledge transfer* mechanism, utilizing critical local layers as the knowledge source and server-generated pseudo-data for privacy-preserving local-to-global knowledge sharing.

- We design *Knowledge Gain Entropy*, a novel metric to quantify knowledge density of global layers relative to local critical layers. KGE guides global-to-local knowledge alignment, enabling personalized knowledge upgrades only for clients with verifiable global gains.

- We provide the rigorous theoretical analysis of FedFree's convergence on both non-convex and strongly convex. Extensive experiments on CIFAR-10 and CIFAR-100 demonstrate that FedFree consistently outperforms eight state-of-the-art homogeneous and heterogeneous FL baselines, achieving significant accuracy improvements across diverse settings.

## 2   Related Work

Existing approaches attempt to enable knowledge sharing across heterogeneous clients primarily through knowledge distillation, representation sharing, and architectural adaptation techniques.

**Knowledge Distillation in HtFL.** Knowledge Distillation (KD) is a widely adopted technique where knowledge from one model (teacher) is transferred to another (student), often via soft labels or intermediate features. Several HtFL methods leverage KD, but they frequently rely on auxiliary data. For instance, FedMD [24] requires a shared proxy dataset for clients to compute and exchange soft predictions. FedDF [25] uses logits averaged over client models on a proxy dataset to create a global ensemble teacher for distillation. FedKD [26] distills knowledge from diverse client models

into a global server model, typically assuming access to server-side or proxy data for the distillation process. FedGKD [5] focuses on mitigating client drift in FL using KD from historical global models, but primarily targets statistically heterogeneous settings with homogeneous models.

While effective in some contexts, the reliance of many KD-based HtFL methods on proxy datasets raises privacy concerns and practical hurdles related to data accessibility and relevance, motivating the need for proxy-data-free alternatives.

**Representation and Prototype Sharing.** Another line of work avoids direct weight aggregation by sharing intermediate representations or prototypes. FedProto [13] has clients upload learned class prototypes instead of model parameters, which the server aggregates. FedTGP [14] extends this by introducing trainable global prototypes and contrastive learning to handle heterogeneity better. FCCL [15] learns generalized representations using a cross-correlation matrix derived from unlabeled proxy data and employs KD to prevent forgetting. These methods reduce communication costs and handle architectural differences to some extent. However, they often focus on aligning output representations (logits or prototypes) rather than enabling fine-grained, layer-wise knowledge exchange, and some still depend on proxy data (*e.g.*, FCCL).

**Architectural Adaptation and Matching.** Techniques also exist to handle differing architectures directly. Model transformation methods convert models into a common format; for example, FedMA [27] uses Bayesian matching to align and average layers across heterogeneous neural networks. Model compression and pruning approaches like HeteroFL [28] train a global super-model on the server and deploy dynamically pruned sub-models to clients based on their resources. Others generate personalized models, like pFedHR [29]. While these methods directly tackle model heterogeneity, layer matching can be complex and computationally expensive, pruning may discard potentially useful knowledge, and personalized model generation might not fully leverage collaborative insights from other clients' structures. Furthermore, ensuring optimal knowledge transfer specifically tailored to each client's needs remains a challenge.

**Positioning FedFree.** While notable progress has been made in HtFL, critical challenges remain. Many existing methods still rely on proxy datasets, raising privacy concerns and limiting generalization. Furthermore, even proxy-data-free approaches often lack mechanisms for fine-grained, targeted knowledge exchange: effectively aggregating knowledge (local-to-global) and distributing it optimally (global-to-local) across structurally diverse clients remains an open problem. These gaps highlight the need for frameworks that operate without proxy data while enabling precise, layer-aware knowledge transfer.

## 3 Methodology

### 3.1 Problem Statement

We consider a heterogeneous federated learning setting with $N$ clients. Each client $i \in \{1, \ldots, N\}$ possesses a private dataset $D_i$ and trains a local model $M_i$ characterized by parameters $\theta_i$. Crucially, both the datasets $D_i$ (non-IID) and the model architectures $M_i$ are allowed to be heterogeneous across clients. This heterogeneity reflects real-world constraints where clients may have different hardware capabilities, data distributions, and task requirements, leading to variations in layer types, layer counts, node counts, etc. The collaborative goal is to optimize the performance of these personalized models by minimizing a global objective function, typically formulated as a weighted average of local losses:

$$\min_{\{\theta_i\}_{i=1}^N} \sum_{i=1}^N p_i F_i(M_i, \theta_i; D_i), \tag{1}$$

where $F_i$ is the local loss function for client $i$ on its data $D_i$, $n_i = |D_i|$ is the size of the local dataset, and $p_i = n_i / \sum_{j=1}^N n_j$ is the weighting factor for client $i$. The primary challenge lies in enabling effective knowledge sharing among the heterogeneous parameters $\{\theta_i\}$ without violating privacy or requiring a shared proxy dataset.

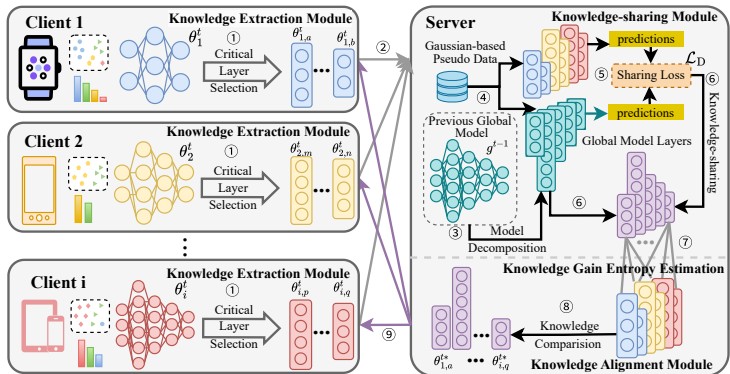

Figure 1: The overview of FedFree architecture. Both $\theta_{1,a}^t, \theta_{1,b}^t, \theta_{2,m}^t, \theta_{2,n}^t, \theta_{i,p}^t, \theta_{i,q}^t$ represent the parameters of the corresponding layers $(a, b, m, n, p, q)$ uploaded by the matching clients $(1, 2, i)$ during the $t$-th round of training.

## 3.2 The FedFree Framework Overview

FedFree addresses the challenges of breaking the knowledge-sharing barriers in HtFL through three specialized modules, (1) **Knowledge Extraction Module**, (2) **Knowledge-Sharing Module**, and (3) **Knowledge Alignment Module** , as depicted in Figure 1. The **Knowledge Extraction Module** operates on the client side, where each client conducts local updates and selectively uploads the most critical layers based on parameter change magnitudes. Subsequently, **Knowledge-Sharing Module** aggregates this knowledge in a reverse layer-wise manner on the server side: it generates Gaussian pseudo-data and updates global model layers by minimizing the output differences with the received critical client layers, thereby avoiding the need for real proxy datasets. Moreover, the **Knowledge Alignment Module** computes the Knowledge Gain Entropy (KGE) for targeted knowledge distribution between the updated global layers and client layers. Finally, it distributes only those global layers offering the richest knowledge improvement back to the respective clients. This targeted distribution ensures architecture-aware personalization without enforcing homogeneous models. Through this design, FedFree enables generalized, scalable, privacy-preserving knowledge transfer across diverse and heterogeneous clients.

## 3.3 Knowledge Extraction Module

This module includes *local update* and *critical layer selection* steps on each participating client. **Local Update.** In round $t$, client $i$ performs standard local training using its private data $D_i$ for $E$ epochs (or steps), updating its local model parameters from $\theta_i^{t-1}$ to $\theta_i^t$. Let $\theta_{i,l}^t$ denote the parameters of the $l$-th layer of client $i$'s model at the end of the local training in round $t$.

**Critical Layer Selection.** To efficiently capture the most significant learning, FedFree identifies critical layers. Client $i$ calculates the magnitude of change for each layer $l$ using the L2-norm:

$$\Delta\theta_{i,l}^t = \frac{\|\theta_{i,l}^t - \theta_{i,l}^{t-1}\|_2^2}{\|\theta_{i,l}^{t-1}\|_2^2}. \tag{2}$$

This $\Delta\theta_{i,l}^t$ reflects the extent of parameter updates for layer $l$. The client then selects a set $L_i^t$ containing the parameters $\{\theta_{i,l}^t\}$ of the top-$k$ layers exhibiting the largest $\Delta\theta_{i,l}^t$. In our experiments, we typically use $k = 2$. These selected critical layer parameters $L_i^t$ are then uploaded to the server. This selective upload reduces communication overhead compared to uploading the entire model.

## 3.4 Knowledge-sharing Module (Local-to-Global)

After critical local knowledge is extracted, the next step is to integrate these insights at the server side without compromising privacy. This server-side module aggregates the knowledge received from clients' critical layers into a global model, crucially, *without using any proxy data*.

**Global Model and Pseudo-data.** The server maintains a global model $G^t$. For the heterogeneous settings evaluated in this work (HtFL), this global model utilizes a specific architecture (detailed in Appendix D.1, Table A1) designed to serve as an effective intermediary for knowledge aggregation and distribution across diverse client models of varying complexities. While its architecture may not be larger than all individual heterogeneous client models, its role is to capture and fuse essential functional knowledge from corresponding client layers via the reverse knowledge transfer mechanism described subsequently. Let $g_j^t$ denote the parameters of the $j$-th layer of this global model at round $t$. Since no real client data is available at the server, and to avoid the complexity of training a separate data generator, we employ a simple yet effective strategy for generating synthetic pseudo-data $\hat{X}$ to probe layer responses. We initialize $\hat{X}$ from a standard multivariate Gaussian distribution, *i.e.*, $\hat{X} \sim \mathcal{N}(0, I)$. This choice provides a diverse set of input signals capable of eliciting varied functional responses from neural network layers without relying on any prior knowledge of the true data distribution or introducing data-dependent privacy risks. The dimensions of $\hat{X}$ are dynamically adjusted based on the input structure of layers. We emphasize that this server-generated $\hat{X}$ contains no client-specific information, thereby preserving privacy.

**Reverse Knowledge Transfer.** The core idea is to update the global model layers $G^t$ such that their outputs on pseudo-data mimic the outputs of the corresponding critical client layers $L^t = \bigcup_i L_i^t$. For each uploaded critical layer $\theta_{i,l}^t \in L^t$, the server identifies a corresponding layer $g_j^{t-1}$ in the previous global model $G^{t-1}$ (typically based on layer type and potentially approximate depth). *Note: Sending the parameters $\theta_{i,l}^t$ is necessary because they are the knowledge we transfer to the global model using $\hat{X}$.* The server feeds the same pseudo-data $\hat{X}$ to both layers to obtain their outputs: $\hat{Y}_{i,l}^t = \theta_{i,l}^t \hat{X}$ and $\hat{Y}_{g_j}^{t-1} = g_j^{t-1} \hat{X}$. The knowledge-sharing process aims to minimize the discrepancy between these outputs using a distillation-like loss, effectively transferring the functional knowledge of $\theta_{i,l}^t$ into $g_j^t$. The loss for updating global layer $g_j$ is computed by averaging over all client layers $\{\theta_{i,l}^t\}$ mapped to it:

$$\mathcal{L}_{\mathrm{D}}(g_j^{t-1}; \{\theta_{i,l}^t\}) = \frac{1}{|\{\theta_{i,l}^t\}|} \sum_{\theta_{i,l}^t} \left\| g_j^{t-1} \hat{X} - \theta_{i,l}^t \hat{X} \right\|_2^2. \tag{3}$$

The global layer $g_j^t$ (weights $w_j^t$ and bias $b_j^t$) are updated using gradient descent on this loss:

$$w_j^t \leftarrow w_j^{t-1} - \alpha \nabla_{w_j^{t-1}} \mathcal{L}_{\mathrm{D}}, \tag{4}$$

$$b_j^t \leftarrow b_j^{t-1} - \alpha \nabla_{b_j^{t-1}} \mathcal{L}_{\mathrm{D}}, \tag{5}$$

where $\alpha$ is the server-side learning rate. This update is performed for all global layers that correspond to received critical client layers, resulting in the updated global model $G^t$.

### 3.5 Knowledge Alignment Module (Global-to-Local)

After aggregating knowledge into the global model $G^t$, this module determines how to best distribute this potentially richer knowledge back to enhance the personalized local models.

**Knowledge Gain Entropy (KGE) Estimation.** To guide the targeted distribution of global knowledge, we introduce the Knowledge Gain Entropy (KGE). KGE is designed as a heuristic metric to assess whether a global model layer $g_j^t$ may offer a richer set of learned patterns compared to a client's local layer $\theta_{i,l}^t$. Our central hypothesis is that a global layer, having been updated using knowledge aggregated from multiple diverse clients, is likely to develop a weight distribution that encodes a broader range of features. We propose that this increased diversity can be reflected in the Shannon entropy $H(\cdot)$ of its quantized weight distribution. Thus, KGE is defined as the difference in entropy:

$$KGE(\theta_{i,l}^t, g_j^t) = H(g_j^t) - H(\theta_{i,l}^t), \quad \text{if type}(\theta_{i,l}^t) = \text{type}(g_j^t). \tag{6}$$

In practice, $H(\cdot)$ is estimated from the flattened weight tensor after quantization into a set number of bins (*e.g.*, 256 bins) [30]. A higher entropy $H(g_j^t)$ is interpreted as suggesting potentially more diverse learned patterns within the global layer.

We acknowledge that relating higher entropy directly to knowledge gain or improved generalization is a guiding principle in FedFree, and that high entropy could, in some isolated instances, indicate

less converged or noisier parameters. However, within our framework, where global layers aggregate diverse information and clients subsequently perform local training, KGE serves as a computationally efficient proxy to identify global layers that are promising candidates for enhancing client models. The intuition is that layers with a positive KGE have assimilated a wider variety of information than the client's current local layer, offering potential for beneficial updates. The empirical success of KGE, as demonstrated in Section 4, supports its utility for targeted knowledge transfer.

**Targeted Knowledge Distribution.** For each client $i$ that uploaded critical layers $L_i^t$, the server identifies the global layer $g_{j*}^t$ which maximizes the positive Knowledge Gain Entropy (KGE) with respect to any of its uploaded critical layers $\theta_{i,l}^t \in L_i^t$. That is, $g_{j*}^t = \arg\max_{g_j^t}\{KGE(\theta_{i,l}^t, g_j^t) > 0 \mid \theta_{i,l}^t \in L_i^t\}$. If such a $g_{j*}^t$ exists (*i.e.*, at least one global layer offers potential knowledge gain), its parameters are sent back to client $i$. The client then *replaces* its corresponding local layer $\theta_{i,l}^t$ with the received global layer $g_{j*}^t$. This ensures that clients receive targeted updates expected to be most beneficial, rather than a simple average. If no global layer yields a positive KGE for a client's critical layers, no layer is sent back to that client for that communication round.

**Handling Dimensional Mismatches.** Dimensional discrepancies between global layer $g_{j*}^t$ and local layer $\theta_{i,l}^t$ are resolved through partial parameter space projection. Let $d_g$ and $d_l$ denote the flattened weight dimensions of global and local layers, respectively, with minimum projection dimension $d_{\min} = \min(d_g, d_l)$. The first $d_{\min}$ elements of global parameters are projected to corresponding positions in the local layer, while exceeding dimensions are zero-initialized to preserve structural consistency:

$$\theta_{i,l}^t[r] = \begin{cases} g_{j*}^t[r], & 1 \le r \le d_{\min}, \\ 0, & \text{otherwise.} \end{cases} \tag{7}$$

This projection maximizes retention of global knowledge in shared parameter space through direct element-wise correspondence ($0 \le r \le d_{\min}$), while unprojected dimensions ($r \ge d_{\min}$) retain local initialization to ensure stable model evolution.

## 3.6 Convergence Analysis

We provide a theoretical analysis of the convergence behavior of FedFree under standard assumptions commonly adopted in federated optimization literature, including L-smoothness, $\mu$-strong convexity, bounded variance, and bounded gradients. These assumptions, along with complete proof details, are presented in Appendix B.

We establish the convergence of FedFree on both the strongly convex and the non-convex settings.

**Theorem 1** (**Strongly Convex Case**). *Suppose Assumptions 1–4 (Appendix B.1) hold, with $\mu > 0$, and let the server and client learning rates $\alpha_t$ and $\eta_t$ be appropriately chosen. Then, after $T$ communication rounds, the iterates generated by FedFree (Algorithm 1) satisfy:*

$$\mathbb{E}\left[F(\theta^T)\right] - F^* = O\left(\frac{1}{T}\right), \tag{8}$$

*where $\theta^T$ denotes the final global model, and $F^*$ is the optimal value of the objective function.*

**Theorem 2** (**Non-Convex Case**). *Under Assumptions 1, 3, and 4 (Appendix B.1) and appropriately chosen learning rates, FedFree achieves the following convergence result in the non-convex setting:*

$$\min_{t\in\{0,...,T-1\}} \mathbb{E}\left[\|\nabla F(\overline{\theta}^t)\|^2\right] = O\left(\frac{1}{T}\right), \tag{9}$$

The proofs leverage standard techniques from federated optimization theory, adapted to account for the pseudo-data-based knowledge aggregation and KGE-guided alignment mechanisms unique to FedFree. The complete derivations are provided in Appendix B.

---

**Algorithm 1** FedFree

---

1: **Server initializes** global model parameters $G^0$.
2: **for** each communication round $t = 1, 2, \ldots, T$ **do**
3:    Select a subset of clients $S_t$ (or all clients by default).
4:    **for** each client $i \in S_t$ **in parallel do**
5:       Perform local training: $\theta_i^t \leftarrow \text{LocalUpdate}(\theta_i^{t-1}, D_i)$.
6:       Compute layer updates $\Delta\theta_{i,l}^t$ (Equation 2), select top-$k$ critical layers $L_i^t$.
7:       Send $L_i^t$ to the server.
8:    **end for**
9:    Server aggregates knowledge:
10:    Generate pseudo-data $\hat{X} \sim \mathcal{N}(0, I)$.
11:    **for** each global layer $g_j^t$ corresponding to $L^t$ **do**
12:       Compute distillation loss $\mathcal{L}_\text{D}$ (Equation 3) and update $g_j^t$ (Equations 4, 5).
13:    **end for**
14:    Server distributes knowledge:
15:    **for** each client $i \in S_t$ **do**
16:       Find global layer $g_{j*}^t$ maximizing positive KGE (Equation 6).
17:       **if** $g_{j*}^t$ exists **then**
18:          Send $g_{j*}^t$ to client $i$ for targeted layer replacement.
19:          Insert adapter if dimensions mismatch (Equation 7).
20:       **end if**
21:    **end for**
22: **end for**

---

## 4 Evaluations

### 4.1 Experimental Setup

**Datasets.** We use two standard benchmark datasets for image classification: CIFAR-10 [31] and CIFAR-100 [31]. These datasets provide diverse class structures suitable for evaluating FL algorithms under different levels of task complexity.

**Statistical Heterogeneity.** To simulate real-world non-IID data distributions across clients, we partition the datasets using a method similar to [32]. Specifically, for non-IID1, in CIFAR-10, we distribute data from 1 out of 10 categories to each client (non-IID: 1/10). In CIFAR-100, we divide the data from 10 out of 100 categories for each client (non-IID: 10/100). For the data distribution bias study (Section 4.3), we also evaluate IID and other non-IID partitioning strategies based on label skew as follows. Details are provided with the specific experiment.

**Model Heterogeneity.** To evaluate performance under model heterogeneity (HtFL settings), we define eight distinct architectures with varying depths and layer widths, randomly assigning one architecture to each client. The specific configurations of these eight distinct CNN architectures, along with the architecture of the server's global model, are detailed in Appendix D.1 (Table A1). For homogeneous FL (HmFL settings) comparisons, all clients use the same standard CNN architecture (also detailed in Appendix D.1).

**Baselines.** We compare FedFree against eight representative FL methods, categorized by their handling of heterogeneity:

- **Homogeneous Baselines (HmFL):** FedAvg [33], Per-FedAvg [34] (personalization via meta-learning), FedProx [35] (handles statistical heterogeneity via regularization), and pFedLA [36] (personalization via layer-wise aggregation). These are evaluated in settings where clients have identical model architectures but non-IID data.

- **Heterogeneous Baselines (HtFL):** FedProto [13] (prototype), FedKD [26] (KD, adapted for comparison), FedGH [37] (shared global head), and FedTGP [14] (trainable global prototypes). These are evaluated in settings where both data and models are heterogeneous.

Table 1: Comparison of final average test accuracy (%). HmFL denotes Homogeneous Models (non-IID1), HtFL denotes Heterogeneous Models (non-IID1). ↑ indicates FedFree's percentage point improvement over the best-performing baseline in that category (HmFL or HtFL).

| Setting | Algorithm | 10 Clients | | 20 Clients | | 100 Clients | |
|---|---|---|---|---|---|---|---|
| | | CIFAR-10 | CIFAR-100 | CIFAR-10 | CIFAR-100 | CIFAR-10 | CIFAR-100 |
| HmFL | FedAvg [33] | 54.44 | 25.39 | 52.20 | 23.93 | 29.21 | 16.87 |
| | Per-FedAvg [34] | 86.34 | 36.01 | 80.96 | 32.95 | 76.77 | 34.31 |
| | FedProx [35] | 73.54 | 42.70 | 73.94 | 40.68 | 73.64 | 35.59 |
| | pFedLA [36] | 90.94 | 38.87 | 88.36 | 33.34 | 16.63 | 26.14 |
| | **FedFree** | **91.45** (0.56%↑) | **59.34** (38.97%↑) | **90.85** (2.82%↑) | **52.91** (30.06%↑) | **86.08** (12.13%↑) | **50.16** (40.94%↑) |
| HtFL | FedProto [13] | 81.68 | 32.88 | 67.30 | 22.90 | 28.03 | 18.98 |
| | FedKD [26] | 85.18 | 39.52 | 88.31 | 39.16 | 76.43 | 33.63 |
| | FedGH [37] | 94.24 | 66.28 | 91.85 | 63.60 | 74.14 | 29.93 |
| | FedTGP [14] | 82.08 | 37.63 | 38.04 | 34.99 | 71.18 | 18.70 |
| | **FedFree** | **95.81** (1.67%↑) | **76.83** (15.92%↑) | **95.35** (3.81%↑) | **70.08** (10.19%↑) | **87.03** (13.87%↑) | **49.20** (46.30%↑) |

We select these baselines to cover diverse strategies, including averaging, personalization, regularization, prototype sharing, and knowledge distillation approaches applicable to homogeneous or heterogeneous settings.

**Implementation Details.** All experiments were implemented using Python 3.11 and PyTorch 1.11, executed on a cluster with Intel Xeon Gold 6126 CPUs and NVIDIA GTX 2080Ti / Tesla T4 GPUs. For local training, we use the SGD optimizer with a learning rate of $\eta = 0.01$ and a batch size of 10 for $E = 5$ local epochs per round. For FedFree, we set the critical layer selection parameter $k = 2$ (based on the sensitivity analysis in Appendix D.3, Figure A1) and the server-side learning rate $\alpha = 0.01$ unless otherwise specified. Total communication rounds $T = 200$. Further details, including specific hyperparameters for baselines (tuned for fair comparison where possible), are provided in Appendix D.2. Unless otherwise specified, results are averaged over 3 runs with different random seeds in different settings. Accuracy reported is the average test accuracy across all participating clients on their local test sets after the final communication round.

## 4.2 Evaluation Results and Analysis

Table 1 shows FedFree consistently outperforms all homogeneous and heterogeneous baselines across both datasets and varying clients' numbers. Notably, the performance gains are particularly substantial on the more complex CIFAR-100 dataset, where FedFree achieves improvements of up to 38.97% (over FedProx, 10 clients, HmFL) and 46.3% (over FedKD, 100 clients, HtFL). This suggests that FedFree's layer-wise knowledge sharing and alignment mechanisms are particularly effective when dealing with finer-grained class distinctions and higher model/data diversity.

The superiority over HmFL baselines like pFedLA highlights the benefit of FedFree's KGE-based alignment compared to simpler layer aggregation strategies, even when models are homogeneous. The significant gains over HtFL baselines like FedGH and FedProto underscore the effectiveness of operating directly on layer parameters (via reverse transfer) combined with targeted alignment, compared to sharing only prototypes or classifier heads, especially without relying on proxy data like many KD methods typically do.

Furthermore, observing the detailed learning curves presented in Appendix D.7 (see Figures A3-A6), FedFree generally exhibits stable convergence. The stability, particularly noticeable in the 100-client HtFL setting (*e.g.*, Figure A6(c)), can be attributed to the selective nature of updates (only critical layers uploaded, only high-KGE layers received) and the supervised knowledge-sharing mechanism using pseudo-data, which likely provides a more consistent aggregation signal compared to direct parameter averaging or prototype matching in highly heterogeneous environments.

## 4.3 Impact of Data Skew on Performance

To assess robustness to varying statistical heterogeneity, we evaluate FedFree on CIFAR-100 with 100 clients under three data distribution settings: IID, non-IID with a partitioning ratio of 8:1:1 (non-IID1) and 6:2:2 (non-IID2), respectively [32]. Figure 2 shows the results. As expected, performance is highest under IID settings. While accuracy decreases under non-IID settings, FedFree maintains

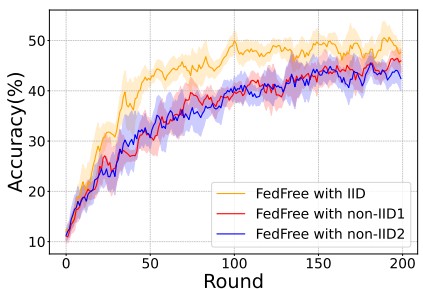

Figure 2: Accuracy on CIFAR-100 with 100 clients across non-IID1 and non-IID2.

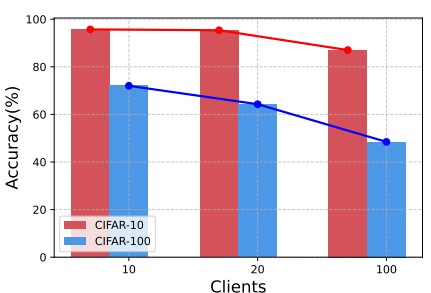

Figure 3: Scalability analysis on the HtFL setting (non-IID1).

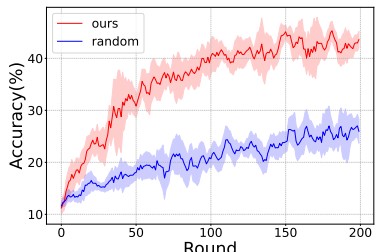

(a) Critical vs. Random Layer Selection

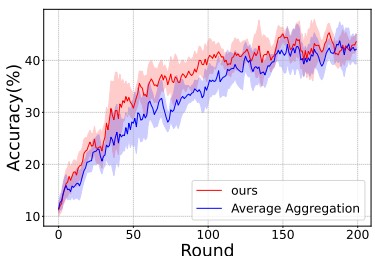

(b) FedFree Sharing vs. Layer Averaging

Figure 4: Ablation analysis on CIFAR-100 with 100 clients (HtFL, non-IID1). Comparing (a) critical layer selection using Equation 2 vs. random layer selection, and (b) FedFree's pseudo-data knowledge sharing vs. simple averaging of uploaded layers.

strong performance even under significant data skew (non-IID1). The performance gap between different levels of non-IID is relatively contained, indicating robustness. This resilience stems from the personalization inherent in the framework: clients primarily update based on local data, and the global model serves to inject useful generalized knowledge identified via KGE, rather than enforcing strict conformity, mitigating the negative impact of conflicting data distributions during aggregation.

## 4.4 Scalability Analysis

We examine how FedFree scales with an increasing number of clients (10, 20, and 100) in the HtFL setting. Figure 3 plots the final accuracy on CIFAR-10 and CIFAR-100. Accuracy moderately decreases as the number of clients grows, particularly from 20 to 100 clients. This trend is common in FL due to the increasing statistical heterogeneity and the potential for more diverse local models as the client pool expands. However, compared with Table 1, FedFree maintains a significant performance advantage over baselines even at 100 clients, demonstrating scalability. The KGE alignment mechanism plays a core role here by selectively incorporating knowledge, potentially preventing the aggregation of conflicting updates from a large, diverse client pool. We observe similar scalability trends under homogeneous model settings as detailed in Figure A2) in Appendix D.5.

## 4.5 Ablation Studies

To validate the contribution of FedFree's core components, we conduct ablation studies on CIFAR-100 with 100 clients in the HtFL setting (non-IID1), shown in Figure 4.

**Critical Layer Selection Strategy.** We compare selecting the top-$k$ layers with the highest change rate (Equation 2, used in FedFree) against randomly selecting $k$ layers to upload (Figure 4(a)). Selecting critical layers based on change rate significantly outperforms random selection. This confirms that the parameter change magnitude effectively identifies layers undergoing significant learn-

ing that are valuable for knowledge sharing. Randomly sharing layers fails to capture the richest updates, leading to poor performance.

**Knowledge-sharing Mechanism.** We compare FedFree's knowledge-sharing aggregation (updating global layers based on minimizing output difference on pseudo-data, Equation 3) against directly averaging the parameters of uploaded critical layers of the same type (ignoring incompatible types). Figure 4(b) shows that FedFree's pseudo-data-based knowledge-sharing significantly outperforms simple layer averaging. This highlights the benefit of using functional knowledge distillation via pseudo-data outputs for aggregation, which is more robust to architectural differences than direct parameter averaging.

These ablations confirm the effectiveness of both the critical layer selection strategy and the pseudo-data-driven knowledge-sharing mechanism within FedFree.

## 5 Conclusion

This paper tackles the challenge of breaking knowledge-sharing barriers in heterogeneous federated learning. We introduced FedFree, a novel framework breaking knowledge-sharing barriers via *reverse layer-wise knowledge transfer* (using pseudo-data for proxy-data-free aggregation) and *Knowledge Gain Entropy* (for targeted, layer-specific knowledge alignment). Supported by theoretical convergence guarantees and extensive experiments, FedFree significantly enhances personalized model performance, consistently outperforming state-of-the-art HtFL baselines on benchmark datasets. Moreover, FedFree offers a practical and privacy-preserving pathway towards realizing the potential of collaborative learning across diverse devices and data distributions. But practical deployment on physical devices and in environments with extremely large models remains untested. In future work, we will evaluate FedFree in real-world deployment scenarios with enhanced heterogeneity, including more diverse model architectures and non-IID data distributions.

**Acknowledgement**

We thank the reviewers for their helpful comments. This work was partially supported by Shanghai Municipal Education Commission Artificial Intelligence Plan (Z2024-119).

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

# Appendix

## A Notation

Table A1 summarizes the main notations used throughout the paper.

Table A1: Notations

| Symbol | Description |
|---|---|
| $N$ | Total number of clients. |
| $T$ | Total number of communication rounds. |
| $E$ | Number of local training epochs/steps per round. |
| $i$ | Index for clients $(1, \ldots, N)$. |
| $t$ | Index for communication rounds $(0, \ldots, T-1)$. |
| $l$ | Index for layers within a model. |
| $\theta_i^t$ | Parameters of the entire local model for client $i$ at the start of round $t$. |
| $\theta_{i,l}^t$ | Parameters of the $l$-th layer for client $i$ at the start of round $t$. |
| $\Delta\theta_{i,l}^t$ | Change in parameters for layer $l$ of client $i$ during round $t$ (Equation 2). |
| $L_i^t$ | Set of critical layer parameters uploaded by client $i$ in round $t$. |
| $k$ | Number of critical layers selected by each client. |
| $G^t$ | Global model parameters at the start of round $t$. |
| $g_j^t$ | Parameters of the $j$-th layer of the global model $G^t$. |
| $d_l$ | Flattened weight dimension of the local layer $\theta_{i,l}^t$ |
| $d_g$ | Flattened weight dimension of the global layer $g_j^t$. |
| $d_{\min}$ | Minimum projection dimension, $d_{\min} = \min(d_g, d_l)$. |
| $\bar{\theta}^t$ | Represents an aggregated state, $e.g.$, $\bar{g}^t = \sum_{i=1}^{N} p_i \nabla F_i(\theta_{i,l}^t)$ (used in proofs). |
| $F_i(\cdot)$ | Local loss function for client $i$. |
| $F(\cdot)$ | Global loss function $F(\theta) = \sum_{i=1}^{N} p_i F_i(\theta)$. |
| $F^*$ | Optimal value of the global loss function. |
| $D_i$ | Local dataset of client $i$. |
| $n_i$ | Size of dataset $D_i$. |
| $p_i$ | Weight of client $i$, $p_i = n_i / \sum_j n_j$. |
| $\hat{X}$ | Server-generated pseudo-data based on $\mathcal{N}(0, I)$. |
| $\mathcal{L}_D$ | Knowledge-sharing loss on the server (Equation 3). |
| $H(\cdot)$ | Entropy of a layer's weight distribution (Equation 6). |
| KGE | Knowledge Gain Entropy (Equation 6). |
| $\tilde{L}_{i,l}^t$ | Adapter layer inserted for client $i$ after layer $l$ in round $t$ (Equation 7). |
| $W, b$ | Parameters of the adapter layer $\tilde{L}_{i,l}^t$. |
| $\eta$ | Learning rate for client-side local training. |
| $\alpha$ | Learning rate for server-side global model updates (Equations 4, 5). |
| $L$ | Lipschitz constant for smoothness (Assumption 1). |
| $\mu$ | Strong convexity constant (Assumption 2). |
| $\sigma_i^2$ | Variance bound for local stochastic gradients (Assumption 3). |
| $G^2$ | Bound for expected squared norm of local stochastic gradients (Assumption 4). |
| $\Gamma$ | Measure of objective function heterogeneity $(F^* - \sum p_i F_i^*)$. |
| $\beta_l$ | Angle between $v_l^t$ and $g_l^t$ (used in proof). |
| $c$ | Constant relating $v_l^t$ and $g_l^t$ (used in proof). |

## B Proof of Convergence Analysis

We present the convergence proof of FedFree under standard assumptions in federated optimization literature. The proof consists of two parts: convergence under strongly convex objectives and non-convex objectives. All necessary assumptions, intermediate lemmas, and detailed derivations are provided below.

## B.1 Assumptions

We first introduce standard assumptions that are used throughout the convergence analysis.

**Assumption 1** (L-smoothness). *Each local objective $F_i$ is L-smooth, meaning that for all $v, w$,*

$$F_i(v) \leq F_i(w) + \langle v - w, \nabla F_i(w) \rangle + \frac{L}{2} \|v - w\|^2, \tag{A1}$$

*where $\langle \cdot, \cdot \rangle$ denotes the standard inner product. Equivalently,*

$$\|\nabla F_i(v) - \nabla F_i(w)\| \leq L \|v - w\|, \tag{A2}$$

*and*

$$F_i^* \leq F_i(w) - \frac{1}{2L} \|\nabla F_i(w)\|^2, \tag{A3}$$

*where $F_i^* = \min_w F_i(w)$.*

**Assumption 2** (Strong Convexity). *Each local objective $F_i$ is $\mu$-strongly convex, i.e., for all $v, w$,*

$$F_i(v) \geq F_i(w) + \langle v - w, \nabla F_i(w) \rangle + \frac{\mu}{2} \|v - w\|^2. \tag{A4}$$

**Assumption 3** (Bounded Variance). *For any client $i$, the variance of the stochastic gradients is bounded:*

$$\mathbb{E} \left[ \|\nabla F_i(\theta_i, \xi_i) - \nabla F_i(\theta_i)\|^2 \right] \leq \sigma_i^2, \tag{A5}$$

*where $\xi_i$ is a random sample from client $i$'s local dataset.*

**Assumption 4** (Bounded Stochastic Gradients). *The second moment of stochastic gradients is uniformly bounded:*

$$\mathbb{E} \left[ \|\nabla F_i(\theta_i, \xi_i)\|^2 \right] \leq G^2. \tag{A6}$$

## B.2 Key Lemmas

We establish several lemmas to facilitate the convergence proofs.

**Lemma 1** (One-Step SGD Progress). *Under Assumptions 1 and 2, for suitable learning rates satisfying $\eta_t + c\alpha_t \cos \beta_l \leq \frac{1}{4L}$, we have:*

$$\mathbb{E} \left[ \|\bar{\theta}_l^{t+1} - \theta_l^*\|^2 \right] \leq (1 - \mu(\eta_t + c\alpha_t \cos \beta_l)) \mathbb{E} \left[ \|\bar{\theta}_l^t - \theta_l^*\|^2 \right] + \eta_t^2 B, \tag{A7}$$

*where $B = \sum_{i=1}^N p_i^2 \sigma_i^2 + 6L\Gamma + 8(E-1)^2 G^2$ and $\Gamma = F^* - \sum_{i=1}^N p_i F_i^* \geq 0$.*

**Lemma 2** (Bounding the Variance). *Under Assumption 3, the variance between stochastic and expected gradients is bounded as:*

$$\mathbb{E} \left[ \|g^t - \bar{g}^t\|^2 \right] \leq \sum_{i=1}^N p_i^2 \sigma_i^2. \tag{A8}$$

**Lemma 3** (Bounding Client-Server Divergence). *Under Assumption 4 and non-increasing learning rates, the expected divergence between local models and the averaged model is bounded as:*

$$\mathbb{E} \left[ \sum_{i=1}^N p_i \|\bar{\theta}_l^t - \theta_{i,l}^t\|^2 \right] \leq 4\eta_t^2 (E-1)^2 G^2. \tag{A9}$$

**Lemma 4** (Gradient Consistency). *Under Assumption 3, the averaged stochastic gradient approximates the true gradient:*

$$\nabla F(\bar{\theta}_l^t) = \sum_{i=1}^N p_i \nabla F_i(\theta_{i,l}^t) = \bar{g}_l^t. \tag{A10}$$

**Lemma 5** (Bounded Global Gradient Norm). *Under Assumption 4, we have:*

$$\mathbb{E} \left[ \|g^t\|^2 \right] \leq \sum_{i=1}^N p_i^2 G^2. \tag{A11}$$

## B.3 Proof of Convergence for Strongly Convex Objectives

We first establish convergence in the strongly convex case.

*Proof of Theorem 1.* The update rule in FedFree is:

$$\bar{\theta}_l^{t+1} = \bar{\theta}_l^t - \eta_t g_l^t - \alpha_t v_l^t, \tag{A12}$$

where $v_l^t$ aligns with $g_l^t$ via $v_l^t = cg_l^t \cos \beta_l$.

Since the directions of $F(\bar{\theta}_l^t)$ and $\nabla \mathcal{L}_{D,i,l}$ are related, so let $v_l^t = cg_l^t cos\beta_l$, where$\beta_l$ is the angle of $v_l^t$ and $g_l^t$.

Let $\Delta_t = \mathbb{E}\|\bar{\theta}_l^t - \theta_l^*\|^2$. From Lemma 1, Lemma 2, Lemma 3, it follows that

$$\Delta_{t+1} \leq (1 - \mu(\eta_t + c\alpha_t cos\beta_l))\Delta_t + \eta_t^2 B, \tag{A13}$$

where

$$B = \sum_{i=1}^{N} p_i^2 \sigma_i^2 + 6L\Gamma + 8(E-1)^2 G^2. \tag{A14}$$

For a diminishing step size, $\eta_t + c\alpha_t cos\beta_l = \frac{\beta}{t+\gamma}$ for some $\beta > \frac{1}{\mu}$ and $\gamma > 0$. We will prove $\Delta_t \leq \frac{v}{\gamma+t}$ where $v = \max\left\{\frac{\beta^2 B}{\beta\mu-1}, (\gamma+1)\Delta_1\right\}$. We prove it by induction. Firstly, the definition of $v$ ensures that it holds for $t = 1$. Assume the conclusion holds for some $t$, it follows that

$$\begin{aligned}
\Delta_{t+1} &\leq (1 - \mu(\eta_t + c\alpha_t cos\beta_l))\Delta_t + (\eta_t + c\alpha_t cos\beta_l)^2 B \\
&\leq \left(1 - \frac{\beta\mu}{t+\gamma}\right)\frac{v}{t+\gamma} + \frac{\beta^2 B}{(t+\gamma)^2} \\
&= \frac{t+\gamma-1}{(t+\gamma)^2}v + \left[\frac{\beta^2 B}{(t+\gamma)^2} - \frac{\beta\mu-1}{(t+\gamma)^2}v\right] \\
&\leq \frac{v}{t+\gamma+1}.
\end{aligned} \tag{A15}$$

Then by the $L$-smoothness of $F(\cdot)$

$$\mathbb{E}[F(\bar{\theta}_l^t)] - F^* \leq \frac{L}{2}\Delta_t \leq \frac{L}{2}\frac{v}{\gamma+t}. \tag{A16}$$

Specially, if we choose $\beta = \frac{2}{\mu}, \gamma = \max\{8\frac{L}{\mu}, E\} - 1$and denote $\kappa = \frac{L}{\mu}$, then $\eta_t + c\alpha_t cos\beta_l = \frac{2}{\mu}\frac{1}{\gamma+t}$. Then, we have

$$\begin{aligned}
v &= \max\left\{\frac{\beta^2 B}{\beta\mu-1}, (\gamma+1)\Delta_1\right\} \leq \frac{\beta^2 B}{\beta\mu-1} + (\gamma+1)\Delta_1 \leq \frac{4B}{\mu^2} + (\gamma+1)\Delta_1, \\
\mathbb{E}[F(\bar{\theta}_l^t)] - F^* &\leq \frac{L}{2}\frac{v}{\gamma+t} \leq \frac{\kappa}{\gamma+t}\left(\frac{2B}{\mu} + \frac{\mu(\gamma+1)}{2}\Delta_1\right).
\end{aligned} \tag{A17}$$

Applying Lemmas 1, 2, and 3, and using a standard induction argument on diminishing step size $\eta_t + c\alpha_t \cos\beta_l = \frac{\beta}{t+\gamma}$ for appropriate $\beta, \gamma$, we derive:

$$\mathbb{E}\left[F(\bar{\theta}_l^t)\right] - F^* = O\left(\frac{1}{t}\right), \tag{A18}$$

where constants depend on $L, \mu, G, E$.

Detailed derivations are provided in the subsequent calculations. □

### B.3.1 Proof of Lemma 1

*Proof.* Note that $\bar{\theta}^{t+1} = \bar{\theta}^t - \eta_t g^t - \alpha_t v^t$.

$$
\begin{aligned}
\left\|\bar{\theta}_l^{t+1} - \theta_l^\star\right\|^2 &= \left\|\bar{\theta}_l^t - (\eta_t + c\alpha_t cos\beta_l)\mathbf{g}_l^t - \theta_l^\star - (\eta_t + c\alpha_t cos\beta_l)\bar{\mathbf{g}}_l^t + (\eta_t + c\alpha_t cos\beta_l)\bar{\mathbf{g}}_l^t\right\|^2 \\
&= \underbrace{\left\|\bar{\theta}_l^t - \theta_l^\star - (\eta_t + c\alpha_t cos\beta_l)\bar{\mathbf{g}}_l^t\right\|^2}_{A_1} \\
&\quad + \underbrace{2(\eta_t + c\alpha_t cos\beta_l)\left\langle\bar{\theta}_l^t - \theta_l^\star - (\eta_t + c\alpha_t cos\beta_l)\bar{\mathbf{g}}_l^t, \bar{\mathbf{g}}_l^t - \mathbf{g}_l^t\right\rangle}_{A_2} \\
&\quad + (\eta_t + c\alpha_t cos\beta_l)^2\left\|\mathbf{g}_l^t - \bar{\mathbf{g}}_l^t\right\|^2.
\end{aligned}
\tag{A19}
$$

Note that $\mathbb{E}A_2 = 0$. We next focus on bounding $A_1$. Again we split $A_1$ into three terms:

$$
\left\|\bar{\theta}_l^t - \theta_l^\star - (\eta_t + c\alpha_t cos\beta_l)\bar{\mathbf{g}}_l^t\right\|^2 = \left\|\bar{\theta}_l^t - \theta_l^\star\right\|^2 \underbrace{-2(\eta_t + c\alpha_t cos\beta_l)\left\langle\bar{\theta}_l^t - \theta_l^\star, \bar{\mathbf{g}}_l^t\right\rangle}_{B_1}
$$
$$
+ \underbrace{(\eta_t + c\alpha_t cos\beta_l)^2\left\|\bar{\mathbf{g}}_l^t\right\|^2}_{B_2}.
\tag{A20}
$$

From the the $L$-smoothness of $F_i(\cdot)$, it follows that

$$
\left\|\nabla F_i\left(\theta_{i,l}^t\right)\right\|^2 \le 2L\left(F_i\left(\theta_{i,l}^t\right) - F_i^\star\right).
\tag{A21}
$$

By the convexity of $\|\cdot\|^2$ and Equation A21, we have

$$
\begin{aligned}
B_2 &= (\eta_t + c\alpha_t cos\beta_l)^2\left\|\bar{\mathbf{g}}^t\right\|^2 \\
&\le (\eta_t + c\alpha_t cos\beta_l)^2\sum_{i=1}^N p_i\left\|\nabla F_i\left(\theta_{i,l}^t\right)\right\|^2 \\
&\le 2L(\eta_t + c\alpha_t cos\beta_l)^2\sum_{i=1}^N p_i\left(F_i(\theta_{i,l}^t) - F_i^*\right).
\end{aligned}
\tag{A22}
$$

Note that

$$
\begin{aligned}
B_1 &= -2(\eta_t + c\alpha_t cos\beta_l)\left\langle\bar{\theta}_l^t - \theta_l^\star, \bar{\mathbf{g}}_l^t\right\rangle \\
&= -2(\eta_t + c\alpha_t cos\beta_l)\sum_{i=1}^N p_i\left\langle\bar{\theta}_l^t - \theta_l^\star, \nabla F_i(\theta_{i,l}^t)\right\rangle \\
&= -2(\eta_t + c\alpha_t cos\beta_l)\sum_{i=1}^N p_i\left\langle\bar{\theta}_l^t - \theta_{i,l}^t, \nabla F_i(\theta_{i,l}^t)\right\rangle \\
&\quad - 2(\eta_t + c\alpha_t cos\beta_l)\sum_{i=1}^N p_i\left\langle\theta_{i,l}^t - \theta_l^\star, \nabla F_i(\theta_{i,l}^t)\right\rangle.
\end{aligned}
\tag{A23}
$$

By Cauchy-Schwarz inequality and AM-GM inequality, we have

$$
-2\left\langle\bar{\theta}_l^t - \theta_{i,l}^t, \nabla F_i\left(\theta_{i,l}^t\right)\right\rangle \le \frac{1}{(\eta_t + c\alpha_t cos\beta_l)}\left\|\bar{\theta}_l^t - \theta_{i,l}^t\right\|^2 + (\eta_t + c\alpha_t cos\beta_l)\left\|\nabla F_i\left(\theta_{i,l}^t\right)\right\|^2.
\tag{A24}
$$

By the $\mu$-strong convexity o $F_i(\cdot)$, we have

$$
-\left\langle\theta_t^k - \theta_l^\star, \nabla F_i\left(\theta_{i,l}^t\right)\right\rangle \le -\left(F_i\left(\theta_{i,l}^t\right) - F_i(\theta_l^*)\right) - \frac{\mu}{2}\left\|\theta_{i,l}^t - \theta_l^\star\right\|^2.
\tag{A25}
$$

By combining the equations above, it follows that

$$A_1 = \left\| \bar{\theta}_l^t - \theta_l^\star - (\eta_t + c\alpha_t cos\beta_l)\bar{\mathbf{g}}^t \right\|^2$$

$$\leq \left\| \bar{\theta}_l^t - \theta_l^\star \right\|^2 + 2L(\eta_t + c\alpha_t cos\beta_l)^2 \sum_{i=1}^{N} p_i \left( F_i(\theta_{i,l}^t) - F_i^* \right)$$

$$+ (\eta_t + c\alpha_t cos\beta_l) \sum_{i=1}^{N} p_i \left( \frac{1}{(\eta_t + c\alpha_t cos\beta_l)} \left\| \bar{\theta}_l^t - \theta_{i,l}^t \right\|^2 + (\eta_t + c\alpha_t cos\beta_l) \left\| \nabla F_i \left( \theta_{i,l}^t \right) \right\|^2 \right)$$

$$- 2(\eta_t + c\alpha_t cos\beta_l) \sum_{i=1}^{N} p_i \left( F_i \left( \theta_{i,l}^t \right) - F_i(\theta^*) + \frac{\mu}{2} \left\| \theta_{i,l}^t - \theta_l^\star \right\|^2 \right)$$

$$= (1 - \mu(\eta_t + c\alpha_t cos\beta_l)) \left\| \bar{\theta}_l^t - \theta_l^\star \right\|^2 + \sum_{i=1}^{N} p_i \left\| \bar{\theta}_l^t - \theta_{i,l}^t \right\|^2$$

$$+ \underbrace{4L(\eta_t + c\alpha_t cos\beta_l)^2 \sum_{i=1}^{N} p_i \left( F_i(\theta_{i,l}^t) - F_i^* \right) - 2(\eta_t + c\alpha_t cos\beta_l) \sum_{i=1}^{N} p_i \left( F_i \left( \theta_{i,l}^t \right) - F_i(\theta_l^*) \right)}_{C}.$$

(A26)

We next aim to bound $C$. We define : $\gamma_t = 2(\eta_t + c\alpha_t cos\beta_l)(1 - 2L(\eta_t + c\alpha_t cos\beta_l))$. Since $(\eta_t + c\alpha_t cos\beta_l) \leq \frac{1}{4L}, (\eta_t + c\alpha_t cos\beta_l) \leq \gamma_t \leq 2(\eta_t + c\alpha_t cos\beta_l)$. Then we split $C$ into two terms:

$$C = -2(\eta_t + c\alpha_t cos\beta_l)(1 - 2L(\eta_t + c\alpha_t cos\beta_l)) \sum_{i=1}^{N} p_i \left( F_i(\theta_{i,l}^t) - F_i^* \right)$$

$$+ 2(\eta_t + c\alpha_t cos\beta_l) \sum_{i=1}^{N} p_i \left( F_i(\theta_l^*) - F_i^* \right)$$

$$= -\gamma_t \sum_{i=1}^{N} p_i \left( F_i(\theta_{i,l}^t) - F^* \right) + (2(\eta_t + c\alpha_t cos\beta_l) - \gamma_t) \sum_{i=1}^{N} p_i \left( F^* - F_i^* \right)$$

(A27)

$$= \underbrace{-\gamma_t \sum_{i=1}^{N} p_i \left( F_i(\theta_{i,l}^t) - F^* \right)}_{D} + 4L(\eta_t + c\alpha_t cos\beta_l)^2 \Gamma,$$

where in the last equation, we use the notation $\Gamma = \sum_{i=1}^{N} p_i(F^* - F_i^*) = F^* - \sum_{i=1}^{N} p_i F_i^*$.
To bound $D$, we have

$$\sum_{i=1}^{N} p_i \left( F_i(\theta_{i,l}^t) - F^* \right) = \sum_{i=1}^{N} p_i \left( F_i(\theta_{i,l}^t) - F_i(\bar{\theta}_l^t) \right) + \sum_{i=1}^{N} p_i \left( F_i(\bar{\theta}_l^t) - F^* \right)$$

$$\geq \sum_{i=1}^{N} p_i \left\langle \nabla F_i(\bar{\theta}_l^t), \theta_{i,l}^t - \bar{\theta}_l^t \right\rangle + (F(\bar{\theta}_l^t) - F^*)$$

$$\geq -\frac{1}{2} \sum_{i=1}^{N} p_i \left[ (\eta_t + c\alpha_t cos\beta_l) \left\| \nabla F_i(\bar{\theta}_l^t) \right\|^2 + \frac{1}{(\eta_t + c\alpha_t cos\beta_l)} \left\| \theta_{i,l}^t - \bar{\theta}_l^t \right\|^2 \right] \quad \text{(A28)}$$

$$+ (F(\bar{\theta}_l^t) - F^*)$$

$$\geq -\sum_{i=1}^{N} p_i \left[ (\eta_t + c\alpha_t cos\beta_l)L \left( F_i(\bar{\theta}_l^t) - F_i^* \right) + \frac{1}{2(\eta_t + c\alpha_t cos\beta_l)} \left\| \theta_{i,l}^t - \bar{\theta}_l^t \right\|^2 \right]$$

$$+ (F(\bar{\theta}_l^t) - F^*),$$

where the first inequality results from the convexity of $F_i(\cdot)$, the second inequality from AM-GM inequality, and the third inequality from Equation A21. Therefore

$$
\begin{aligned}
\mathrm{C} &= \gamma_t \sum_{i=1}^{N} p_i \left[ (\eta_t + c\alpha_t cos\beta_l)L \left( F_i(\bar{\theta}_l^t) - F_i^* \right) + \frac{1}{2(\eta_t + c\alpha_t cos\beta_l)} \left\| \theta_{i,l}^t - \bar{\theta}_l^t \right\|^2 \right] \\
&\quad - \gamma_t \left( F(\bar{\theta}_l^t) - F^* \right) + 4L(\eta_t + c\alpha_t cos\beta_l)^2 \Gamma \\
&= \gamma_t((\eta_t + c\alpha_t cos\beta_l)L - 1) \sum_{i=1}^{N} p_i \left( F_i(\bar{\theta}_l^t) - F^* \right) \\
&\quad + \left( 4L(\eta_t + c\alpha_t cos\beta_l)^2 + \gamma_t(\eta_t + c\alpha_t cos\beta_l)L \right) \Gamma \\
&\quad + \frac{\gamma_t}{2(\eta_t + c\alpha_t cos\beta_l)} \sum_{i=1}^{N} p_i \left\| \theta_{i,l}^t - \bar{\theta}_l^t \right\|^2 \\
&\leq 6L(\eta_t + c\alpha_t cos\beta_l)^2 \Gamma + \sum_{i=1}^{N} p_i \left\| \theta_{i,l}^t - \bar{\theta}_l^t \right\|^2,
\end{aligned}
$$
(A29)

where in the last inequality, we use the following facts: (1) $(\eta_t + c\alpha_t cos\beta_l)L - 1 \leq -\frac{3}{4} \leq 0$ and $\sum_{i=1}^{N} p_i \left( F_i(\bar{\theta}_l^t) - F^* \right) = F(\bar{\theta}_l^t) - F^* \geq 0$, (2) $\Gamma \geq 0$, and $4L(\eta_t + c\alpha_t cos\beta_l)^2 + \gamma_t(\eta_t + c\hat{\alpha}_t cos\beta_l)L \leq 6(\eta_t + c\hat{\alpha}_t cos\beta_l)^2 L$ and (3) $\frac{\gamma_t}{2(\eta_t + c\alpha_t cos\beta_l)} \leq 1$.

Recalling the expression of $A_1$ and plugging $C$ into it, we have

$$
\begin{aligned}
A_1 &= \left\| \bar{\theta}_l^t - \theta^\star - (\eta_t + c\alpha_t cos\beta_l)\bar{\mathbf{g}}^t \right\|^2 \\
&\leq (1 - \mu(\eta_t + c\alpha_t cos\beta_l)) \left\| \bar{\theta}_l^t - \theta_l^\star \right\|^2 + 2 \sum_{i=1}^{N} p_i \left\| \bar{\theta}_l^t - \theta_{i,l}^t \right\|^2 \\
&\quad + 6(\eta_t + c\alpha_t cos\beta_l)^2 L\Gamma.
\end{aligned}
$$
(A30)

$\square$

### B.3.2 Proof of Lemma 2

*Proof.* From Assumption 3, the variance of the stochastic gradients in device $k$ is bounded.

$$
\begin{aligned}
\mathbb{E} \left\| \mathbf{g}^t - \bar{\mathbf{g}}^t \right\|^2 &= \mathbb{E} \left\| \sum_{i=1}^{N} p_i (\nabla F_i(\theta_{i,l}^t, \xi_i^t)) - \nabla F_i(\theta_{i,l}^t)) \right\|^2 \\
&= \sum_{i=1}^{N} p_i^2 \mathbb{E} \left\| \nabla F_i(\theta_{i,l}^t, \xi_i^t) - \nabla F_i(\theta_{i,l}^t) \right\|^2 \\
&\leq \sum_{i=1}^{N} p_i^2 \sigma_i^2.
\end{aligned}
$$
(A31)

$\square$

### B.3.3 Proof of Lemma 3

*Proof.*

$$\mathbb{E}\sum_{i=1}^{N}p_i\left\|\bar{\theta}_l^t-\theta_{i,l}^t\right\|^2 = \mathbb{E}\sum_{i=1}^{N}p_i\left\|(\theta_{i,l}^t-\bar{\theta}^{t_0})-(\bar{\theta}_l^t-\bar{\theta}^{t_0})\right\|^2$$

$$\leq \mathbb{E}\sum_{i=1}^{N}p_i\left\|\theta_i^t-\bar{\theta}^{t_0}\right\|^2$$

$$\leq \sum_{i=1}^{N}p_i\mathbb{E}\sum_{t=t_0}^{t-1}(E-1)(\eta_t+c\alpha_t cos\beta_l)^2\left\|\nabla F_i(\theta_{i,l}^t,\xi_i^t)\right\|^2 \tag{A32}$$

$$\leq \sum_{i=1}^{N}p_i\sum_{t=t_0}^{t-1}(E-1)(\eta_{t_0}+c\alpha_{t_0}cos\beta_l)^2 G^2$$

$$\leq \sum_{i=1}^{N}p_i(\eta_{t_0}+c\alpha_{t_0}cos\beta_l)^2(E-1)^2 G^2$$

$$\leq 4(\eta_t+c\alpha_t cos\beta_l)^2(E-1)^2 G^2.$$

Here in the first inequality, we use $\mathbb{E}\|X-\mathbb{E}X\|^2 \leq \mathbb{E}\|X\|^2$ with probability $p_i$. In the second inequality, we use Jensen inequality:

$$\left\|\theta_{i,l}^t-\bar{\theta}^{t_0}\right\|^2 = \left\|\sum_{t=t_0}^{t-1}(\eta_t+c\alpha_t cos\beta_l)\nabla F_i(\theta_{i,l}^t,\xi_i^t)\right\|^2 \leq (t-t_0)\sum_{t=t_0}^{t-1}(\eta_t+c\alpha_t cos\beta_l)^2\left\|\nabla F_i(\theta_{i,l}^t,\xi_i^t)\right\|^2. \tag{A33}$$

$\square$

## B.4 Proof of Convergence for Non-Convex Objectives

We next establish convergence under non-convex settings.

*Proof of Theorem 2.* Using $L$-smoothness (Assumption 1), we have for the update:

$$F(\bar{\theta}_l^{t+1}) \leq F(\bar{\theta}_l^t) + \langle\nabla F(\bar{\theta}_l^t),\bar{\theta}_l^{t+1}-\bar{\theta}_l^t\rangle + \frac{L}{2}\|\bar{\theta}_l^{t+1}-\bar{\theta}_l^t\|^2. \tag{A34}$$

Since the directions of $F(\bar{\theta}_l^t)$ and $\nabla\mathcal{L}_{D,k,l}$ are related, so let $v_l^t = cg_l^t cos\beta_l$, where $\beta_l$ is the angle of $v_l^t$ and $g_l^t$.

$$F(\bar{\theta}_l^{t+1}) - F(\bar{\theta}_l^t) + (\eta_t+c\alpha_t)\langle\nabla F(\bar{\theta}_l^t),g_l^t\rangle \leq \frac{L}{2}(g_l^t+c\alpha_t cos\beta_l)^2\|g_l^t\|^2. \tag{A35}$$

We know $\|\mathbf{a}+\mathbf{b}\|^2 = \|\mathbf{a}\|^2+2<\mathbf{a},\mathbf{b}>+\|\mathbf{b}\|^2$. Thus, $<\mathbf{a},\mathbf{b}> = \frac{1}{2}\|\mathbf{a}\|^2+\frac{1}{2}\|\mathbf{b}\|^2-\frac{1}{2}\|\mathbf{a}-\mathbf{b}\|^2$. So let $\langle\nabla F(\bar{\theta}_l^t),g_l^t\rangle$:

$$\langle\nabla F(\bar{\theta}_l^t),g_l^t\rangle = \frac{1}{2}\|\nabla F(\bar{\theta}_l^t)\|^2+\frac{1}{2}\|g_l^t\|^2-\frac{1}{2}\|\nabla F(\bar{\theta}_l^t)-g_l^t\|^2. \tag{A36}$$

Substitute Equation A36 into Equation A35, we get

$$F(\bar{\theta}_l^{t+1})-F(\bar{\theta}_l^t)+\frac{\eta_t+c\alpha_t cos\beta_l}{2}(\|\nabla F(\bar{\theta}_l^t)\|^2+\|g_l^t\|^2-\|\nabla F(\bar{\theta}_l^t)-g_l^t\|^2) \leq \frac{L(\eta+c\alpha_t cos\beta_l)^2}{2}\|g_l^t\|^2. \tag{A37}$$

Multiply both sides of Equation A37 by $\frac{\eta_t+c\alpha_t cos\beta_l}{2}$, we have

$$\frac{\eta_t+c\alpha_t cos\beta_l}{2}[F(\bar{\theta}_l^{t+1})-F(\bar{\theta}_l^t)]+\|\nabla F(\bar{\theta}_l^t)\|^2+\|g_l^t\|^2-\|\nabla F(\bar{\theta}_l^t)-g_l^t\|^2 \leq L(\eta+c\alpha_t cos\beta_l)\|g_l^t\|^2. \tag{A38}$$

So we get

$$\|\nabla F(\bar{\theta}_l^t)\|^2 \leq \frac{-2}{\eta_t + c\alpha_t cos\beta_l}[F(\bar{\theta}_l^{t+1}) - F(\bar{\theta}_l^t)] + \|\nabla F(\bar{\theta}_l^t) - g_l^t\|^2 + (L\eta_t + Lc\alpha_t cos\beta_l - 1)\|g_l^t\|^2.$$

(A39)

Combine Lemma 2, Lemma 4, Lemma 5, we have

$$\mathbb{E}\|\nabla F(\bar{\theta}_l^t)\|^2 \leq \frac{2}{\eta_t + c\alpha_t cos\beta_l}\mathbb{E}[F(\bar{\theta}_l^{t+1}) - F(\bar{\theta}_l^t)] + \sum_{i=1}^{N} p_i^2\sigma_i^2 + (L\eta_t + Lc\alpha_t cos\beta_l - 1)p_i^2 G^2).$$

(A40)

Summing over $t \in \{0, 1, \cdots, T-1\}$ and dividing both sides by T, we have

$$\mathbb{E}\|\nabla F(\bar{\theta}_l^t)\|^2 \leq \frac{2}{T(\eta_t + c\alpha_t cos\beta_l)}\mathbb{E}[F(\bar{\theta}^0) - F^*] + \sum_{i=1}^{N} p_i^2\sigma_i^2 + (L\eta_t + Lc\alpha_t cos\beta_l - 1)p_i^2 G^2).$$

(A41)

Substituting the update rule and rearranging terms, combined with Lemmas 2 and 5, yields:

$$\min_{t\in\{0,...,T-1\}} \mathbb{E}\left[\|\nabla F(\bar{\theta}_l^t)\|^2\right] = O\left(\frac{1}{T}\right).$$

(A42)

$\square$

## B.5   Conclusion

Thus, we complete the convergence analysis for FedFree, establishing $O(1/T)$ rates under both strongly convex and non-convex settings.

## C   Discussion

In this work, we address the obstacle in knowledge transfer for heterogeneous federated learning: due to significant variations in device capabilities, knowledge encoded in different model architectures struggles to be effectively aggregated and disseminated across the federation. To tackle this issue, we introduce FedFree, a proxy-data-free HtFL framework that proposes a reverse layer-wise knowledge transfer mechanism, aggregating heterogeneous client models into a global model solely using Gaussian-based pseudo-data, thereby eliminating reliance on real proxy datasets. Fed-Free leverages Knowledge Gain Entropy (KGE) to guide targeted layer-wise knowledge alignment, ensuring each client receives the most beneficial and architecture-specific global updates. The effectiveness of our approach is supported by theoretical analysis and extensive experiments conducted on various datasets and models for computer vision tasks.

Limitations remain, particularly regarding real-world applicability. While FedFree has demonstrated effectiveness in near-real-world settings, its physical deployment on actual devices and environments with extremely large models remains untested due to resource constraints. Real-world implementation of FedFree may reveal additional challenges or limitations, providing further insights into its scalability and efficiency.

## D   Supplementary Experimental Details and Results

This section provides additional details referenced in Section 4.

### D.1   Model Architectures

Table A1 details the architectures used in the HtFL experiments. "Homo. Model" refers to the standard architecture used in homogeneous settings. "Model_1" through "Model_8" represent the eight distinct architectures randomly assigned to clients in the HtFL setting. The "Global Model" architecture used on the server is also shown; it is designed to be encompassing enough for the client models.

Table A1: Model Configurations Comparison (Homogeneous and Heterogeneous Settings).

| Name | Base Architecture | Key Features | Layers | Params. |
|---|---|---|---|---|
| Model_1 | CNN | 2 Conv layers, 3 FC layers | 5 | ≈3.2M |
| Model_2 | ResNet18 | BasicBlock, [2,2,2,2] layers | 18 | ≈11.2M |
| Model_3 | ResNet34 | BasicBlock, [3,4,6,3] layers | 34 | ≈21.3M |
| Model_4 | ResNet50 | Bottleneck, [3,4,6,3] layers | 50 | ≈23.5M |
| Model_5 | ResNet101 | Bottleneck, [3,4,23,3] layers | 101 | ≈42.5M |
| Model_6 | ResNet152 | Bottleneck, [3,8,36,3] layers | 152 | ≈58.2M |
| Model_7 | GoogleNet | Inception modules | 22 | ≈6.8M |
| Model_8 | MobileNetV2 | Linear Bottlenecks, Depthwise Separable Conv | 53 | ≈3.5M |
| Global Model | CNN | 2 Conv layers, 3 FC layers | 5 | ≈3.2M |
| Homo. Model | CNN | 2 Conv layers, 3 FC layers | 5 | ≈3.2M |

## D.2 Hyperparameter Settings

Table A2 details the hyperparameters used for FedFree and the baseline methods. Parameters for baselines were set based on their original papers or tuned slightly for fair comparison in our experimental environment.

Table A2: Hyperparameter settings for FedFree and baseline methods.

| Parameter | FedFree | FedAvg | Per-FedAvg | FedProx | pFedLA | FedProto | FedKD | FedGH | FedTGP |
|---|---|---|---|---|---|---|---|---|---|
| Local epochs $E$ | 30 | 30 | 30 | 30 | 30 | 30 | 30 | 30 | 30 |
| Local learning rate $\eta$ | 0.01 | 0.01 | 0.01 | 0.01 | 0.01 | 0.01 | 0.01 | 0.01 | 0.01 |
| Server learning rate $\alpha$ | 0.01 | - | - | - | - | - | - | 0.01 | 0.01 |
| Critical layers $k$ | 2 | - | - | - | - | - | - | - | - |
| KGE entropy bins | 256 | - | - | - | - | - | - | - | - |
| Proximal term $\mu$ | - | - | - | 0.01 | - | - | - | - | - |
| Meta step size $\beta$ | - | - | 0.01 | - | - | - | - | - | - |
| Hypernetwork LR $\eta_h$ | - | - | - | - | 0.005 | - | - | - | - |
| Prototype LR $\eta_p$ | - | - | - | - | - | 0.1 | - | - | - |
| Regularisation $\lambda$ | - | - | - | - | - | 10 | - | - | - |
| Distillation temp. $\tau$ | - | - | - | - | - | - | [0.95,0.98] | - | - |
| Contrastive temp. $\tau_c$ | - | - | - | - | - | - | - | - | 100 |
| Optimizer | SGD | SGD | SGD | SGD | SGD | SGD | SGD | SGD | SGD |
| Batch size (local) | 10 | 10 | 10 | 10 | 10 | 10 | 10 | 10 | 10 |

## D.3 Parameter Sensitivity Study (Impact of $k$)

We investigate the impact of the number of critical layers $k$ selected for upload on the performance of FedFree. Experiments were conducted on CIFAR-100 with 100 clients (HtFL setting, non-IID1), varying $k \in \{1, 2, 3\}$. Figure A1 shows the mean accuracy statistics after model convergence.

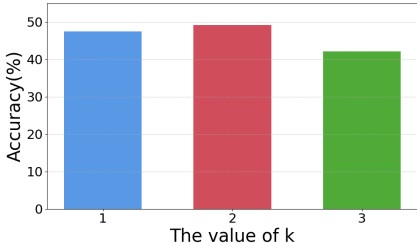

Figure A1: Performance comparison for different values of $k$ (number of critical layers uploaded) on CIFAR-100, 100 clients, HtFL setting.

The results indicate that $k = 2$ achieves the highest mean accuracy, followed by $k = 1$. This suggests that uploading two critical layers optimally captures effective information for knowledge sharing in this setup, while uploading more layers might introduce redundancy or noise without proportional benefits, potentially due to the heterogeneity and resource constraints. Therefore, we used $k = 2$ as the default setting in our experiments.

### D.4 Parameter Sensitivity Study (Entropy Methods)

Based on the results in Table A3, our proposed entropy method achieves the highest accuracy of 49.20% on CIFAR-100, outperforming both Tsallis Entropy (45.98%) and Collision Entropy (44.46%). The superior performance of Shannon entropy can be attributed to its fundamental property as a measure of information content. Unlike Tsallis or Collision entropy, which can be biased towards the most probable events (effectively smoothing the distribution), Shannon entropy is exquisitely sensitive to the entire probability distribution. This demonstrates the superior effectiveness of our approach in handling predictive uncertainty for client selection in federated learning.

Table A3: Comparison of Different Entropy Methods (CIFAR-100, 100 clients.

| Entropy Type | Ours | Tsallis Entropy | Collision Entropy |
|---|---|---|---|
| Accuracy (%) | 49.20 | 45.98 | 44.46 |

### D.5 Additional Scalability Study (Homogeneous Models)

To isolate the effect of client numbers from model heterogeneity, we conducted scalability experiments under a homogeneous model setting (all clients use "Homo. Model" from Table A1).

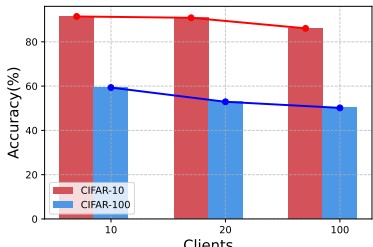

Figure A2: Accuracy variation with different numbers of clients in the HmFL setting (non-IID1).

Figure A2 shows the results for 10, 20, and 100 clients. Similar to the heterogeneous case, accuracy decreases slightly as the number of clients increases, primarily due to escalating statistical heterogeneity. However, FedFree maintains high accuracy compared to the HmFL baselines in Table 1, confirming its scalability even without model heterogeneity.

### D.6 Additional Scalability Study (Additional Dateset)

Table A4: TinyImageNet Benchmark (100 clients).

| Method | Ours | FedProto | FedKD | FedGH | FedTGP |
|---|---|---|---|---|---|
| Accuracy (%) | 23.31 | 16.67 | 13.28 | 17.72 | 15.35 |
| Memory(MB) | 2,045 | 373 | 2,641 | 2,022 | 4,167 |

To further demonstrate the applicability of FedFree, we have conducted additional experiments on the TinyImageNet dataset in Table A4, which is significantly more challenging due to its higher resolution, increased number of classes (200), and greater data complexity.

The results, presented below, demonstrate that FedFree consistently outperforms baseline methods on this more complex dataset, further validating its generalizability. We emphasize that FedFree is model-agnostic and data-agnostic by design, relying on a layer-wise knowledge-sharing strategy that does not assume any specific model architecture or input modality.

### D.7 Additional Learning Curves

This subsection presents detailed learning curves corresponding to the main experimental results summarized in Table 1. These plots illustrate the convergence behavior, specifically the average

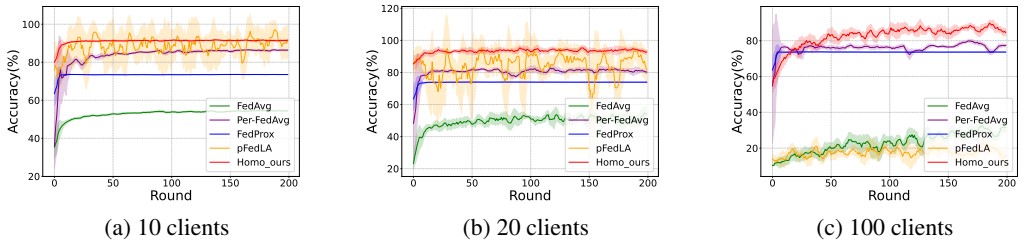

(a) 10 clients        (b) 20 clients        (c) 100 clients

Figure A3: Learning curves (Average Test Accuracy vs. Rounds) for Homogeneous Models (HmFL) on CIFAR-10 (non-IID1). Comparing FedFree against FedAvg, Per-FedAvg, FedProx, and pFedLA for different numbers of clients.

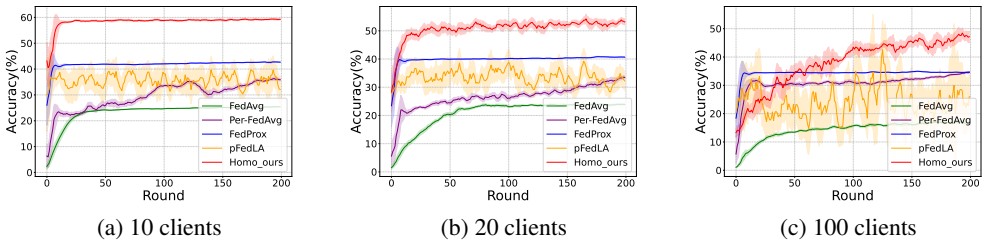

(a) 10 clients        (b) 20 clients        (c) 100 clients

Figure A4: Learning curves (Average Test Accuracy vs. Rounds) for Homogeneous Models (HmFL) on CIFAR-100 (non-IID1). Comparing FedFree against FedAvg, Per-FedAvg, FedProx, and pFedLA for different numbers of clients.

client test accuracy versus communication rounds, for FedFree compared against the relevant baselines under different settings.

**Homogeneous Model Settings (HmFL).** Figures A3 and A4 show the learning curves for the homogeneous model setting (HmFL), where all clients use the same CNN architecture but operate on non-IID data (non-IID1). Comparisons are shown for 10, 20, and 100 participating clients on CIFAR-10 and CIFAR-100, respectively. Baselines included are FedAvg, Per-FedAvg, FedProx, and pFedLA. Shaded areas typically represent the standard deviation across clients or runs.

**Heterogeneous Model Settings (HtFL).** Figures A5 and A6 show the learning curves for the heterogeneous model setting (HtFL), where clients possess different architectures (randomly assigned from Model_1-Model_8 in Table A1) and operate on non-IID data (non-IID1). Comparisons are shown for 10, 20, and 100 participating clients on CIFAR-10 and CIFAR-100, respectively. Baselines included are FedProto, FedKD, FedGH, and FedTGP.

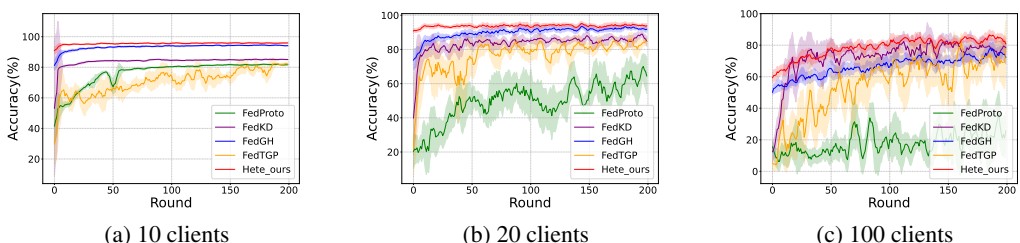

|                    |                   |                    |
| :----------------: | :---------------: | :----------------: |
| (a) 10 clients     | (b) 20 clients    | (c) 100 clients    |

Figure A5: Learning curves (Average Test Accuracy vs. Rounds) for Heterogeneous Models (HtFL) on CIFAR-10 (non-IID1). Comparing FedFree against FedProto, FedKD, FedGH, and FedTGP for different numbers of clients.

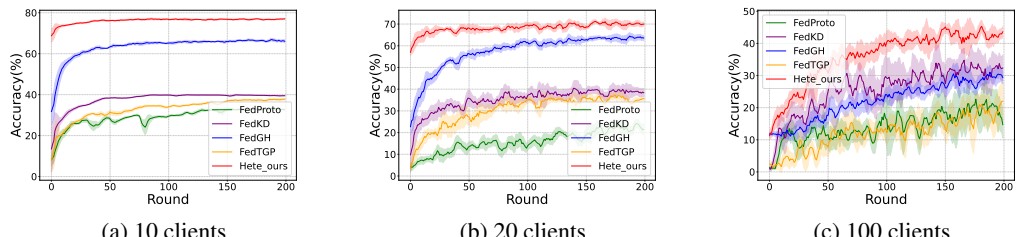

|                    |                   |                    |
| :----------------: | :---------------: | :----------------: |
| (a) 10 clients     | (b) 20 clients    | (c) 100 clients    |

Figure A6: Learning curves (Average Test Accuracy vs. Rounds) for Heterogeneous Models (HtFL) on CIFAR-100 (non-IID1). Comparing FedFree against FedProto, FedKD, FedGH, and FedTGP for different numbers of clients.

