# OpenReview forum: "FedFree: Breaking Knowledge-sharing Barriers through Layer-wise Alignment in Heterogeneous Federated Learning"
_NeurIPS.cc/2025/Conference — NeurIPS 2025 poster_

### Official Review · Reviewer_BXtX · 2025-06-25

**Clarity:** 3
**Significance:** 2
**Originality:** 3
**Rating:** 4
**Confidence:** 4

**Summary:**

This paper proposes FedFree, a model-heterogeneous federated learning method based on knowledge transfer without shared data. FedFree introduces a reverse layer-wise knowledge transfer mechanism and a Knowledge Gain Entropy metric for data-free knowledge transfer between clients and servers. Theoretical analyses and experimental results validate the effectiveness of FedFree.

**Questions:**

Please see  Strengths and Weaknesses.

**Ethical Concerns:**

["NO or VERY MINOR ethics concerns only"]

**Final Justification:**

Thank you for the authors' responses. My concerns have been mostly addressed, and I decide to increase my score to 4.

**Limitations:**

Please see  Strengths and Weaknesses.

**Quality:**

2

**Strengths And Weaknesses:**

Strengths:
1. The paper addresses an important and timely problem in FL. Enabling model-heterogeneous FL has the potential to greatly improve the applicability and flexibility of FL in real-world scenarios.
2. The motivation is clear, and the overall writing quality is good, making the manuscript easy to follow.
3. The experimental setup appears to be highly reproducible, with hyperparameters and implementation details well documented.

Weaknesses:
1. The proposed "Critical Layer Selection" seems to essentially perform layer-wise top-$k$ sparsification. This makes me wonder: since FedFree is motivated by enabling model-heterogeneous FL, what is the rationale for incorporating communication-efficient techniques? Why not simply transferring all layers (rather than only the top-$k$)?
2. In the knowledge distillation stage, the method samples $\hat{X}$ from a standard multivariate Gaussian distribution. Does this implicitly assume that the aggregate data distribution across all clients is well-approximated by a standard Gaussian? If not, how does the method perform if this assumption is violated?
3. What is the computational complexity of the "Reverse Knowledge Transfer" step? Is it $O(L^2)$, where $L$ is the number of layers?
4. The layer-wise distillation strategy in "Reverse Knowledge Transfer" could potentially cause the "short-sight" issue, where each individual layer is optimized without full awareness of the global objective. What are the authors' thoughts or potential remedies regarding this issue?
5. In Line 194, can the authors empirically or theoretically validate the central hypothesis of the method?
6. The experiments are only conducted on CIFAR-10 and CIFAR-100 datasets with various CNN architectures. This experimental evaluation seems limited and may not fully capture the generality or robustness of the proposed approach.
7. In Algorithm 1, Line 12, how many iterations are required to optimize $g_i^t$? What criterion is used for termination of this optimization process?
8. Why is FedFree not compared to methods based on "Architectural Adaptation and Matching," which are also introduced in the related work section?
9. In Figure 4(b), it appears that FedFree and FedAvg achieve similar final convergence (though FedFree converges faster). Given the additional computational overhead in FedFree, it is questionable whether the proposed aggregation method offers a significant advantage over FedAvg.

---

> ### Author Rebuttal · Authors · 2025-07-31
>
> Thanks for your comments.
>
> **Weakness 1 \& Question (Critical Layer Selection/Sparsification)**
>
> **Re**: We appreciate this important question about the rationale behind critical layer selection. The selection of k critical layers (line 153) is motivated by two primary objectives, not solely communication efficiency:
> 1) Effective Knowledge Extraction (Primary): Our core motivation is to capture the most significant learning updates from diverse client models (lines 149--154). The L2-norm of parameter change ($\Delta\Theta$, Equation 2) serves as a robust indicator of which layers have undergone the most substantial and meaningful learning in a given round. By focusing on these "critical layers," FedFree ensures that the most impactful knowledge for the global model is extracted, rather than transmitting redundant or less relevant information. In HtFL, directly aggregating all layers is often impossible or less effective due to architectural mismatches, and transferring all local layers indiscriminately would dilute global learning by introducing uninformative or non-transferable updates. The top-$k$ strategy naturally enforces this selectivity.
> 2) Communication Efficiency (Beneficial Side Effect): While not the sole driver, the resulting reduction in communication overhead by transferring only $k$ layers (line 155) is indeed a significant practical benefit for resource-constrained FL environments.
>
> Therefore, critical layer selection is fundamentally about identifying and transferring **high-value knowledge** in a targeted manner, which inherently aids in both effective HtFL and communication efficiency. The ablation study in Figure 4(a) empirically validates this, showing that selecting critical layers based on parameter change significantly outperforms random selection.
>
> **Weakness 2 \& Question (Gaussian Pseudo-data Assumption)*
>
> **Re**: No, our method **does not assume that the true client data distribution—or the aggregate thereof—is Gaussian**. The use of a standard multivariate Gaussian distribution (X ~ N(0, I), lines 168-174) in FedFree serves a fundamentally different purpose:
>
> 1) Neutral Probing Input: The Gaussian pseudo-data acts as a neutral probing input, allowing us to extract functional responses from both global and local critical layers under the same synthetic stimuli. Its purpose is to provide a "diverse set of input signals capable of eliciting varied functional responses from neural network layers without relying on any prior knowledge of the true data distribution or introducing data-dependent privacy risks'' (lines 170--172).
> 2) Functional Alignment: The goal is to update global layers so that their outputs align with the functional behavior of local layers (i.e., reverse knowledge transfer), given a shared, architecture-agnostic input space. Because the pseudo-data is not required to resemble real data, it avoids privacy risks while still enabling meaningful activation-space alignment.
>
> The empirical success of FedFree (Section 4, particularly Figure 4(b)'s ablation study) demonstrates that this simple, knowledge-agnostic probing mechanism is highly effective in facilitating functional alignment and knowledge transfer even in complex, non-IID settings.
>
> **Weakness 3 \& Question (Computational Complexity of "Reverse Knowledge Transfer")**
>
> **Re**: The computational complexity of the "Reverse Knowledge Transfer" step (Knowledge-sharing Module, Section 3.4) at the server is **not O(N*L**), where N is the number of layers. It is significantly more efficient due to its layer-wise and selective nature.
>
> For each communication round:
>
> 1) Client-side: Clients perform local training (standard FL cost) and then calculate L2-norms of parameter changes for all their layers (negligible). They upload only k critical layers (typically k=2).
> 2) Server-side: Operations include generating Gaussian pseudo-data (negligible cost). For each participating client (denote $C$ as the number of clients) and for each of their $k$ uploaded critical layers, the server identifies a corresponding global layer (an $\mathcal{O}(1)$ lookup), feeds pseudo-data through these two specific layers (one client layer and one global layer), and calculates the distillation loss (Equation 3). The global layer $g_j^t$ is updated in a **single gradient descent step** (Equations~4 and~5) using the server-side learning rate $\alpha$. This cost is proportional to $C \times k \times$ (computation per layer forward pass and loss), which is far less than processing all layers of all client models.
>
> Therefore, the server-side computation is proportional to the number of participating clients multiplied by the number of critical layers (k=2) and the size of those specific layers, making it highly efficient. We will add a dedicated subsection in the Appendix (e.g., D.6) to provide a detailed computational complexity analysis.
>
>
>
> **Weakness 4 \& Question ("Short-sight" issue in Layer-wise Distillation)**
>
> **Re**: We thank the reviewer for highlighting this important concern. While layer-wise distillation might theoretically introduce a "short-sight" issue, FedFree mitigates this through several mechanisms:
>
> 1) Global Model as an Integrating Intermediary: The global model (lines 161-165) integrates knowledge from diverse client-critical layers into a more holistic representation. The aggregation process (Equation 3) considers the functional response of the global layer to all critical client layers mapped to it, encouraging a generalized understanding.
> 2) Client-Side Full Model Training and Adaptation: After receiving an updated global layer, the client model continues its **local training on its full private dataset and entire model architecture** (Algorithm 1, line 5). This allows the client to integrate the global knowledge into its complete model structure, adapt it to its specific data, and ensure consistency with its overall local objective.
> 3) KGE-Guided Beneficial Transfer: The Knowledge Gain Entropy (KGE) mechanism (lines 210-214) explicitly ensures clients only receive global layers that offer positive knowledge gain, preventing the injection of potentially detrimental updates.
>
> These mechanisms ensure that while knowledge is transferred layer-wise, it is ultimately integrated into and validated by the full model's local training, mitigating the "short-sight" issue.
>
>
> **Weakness 5 \& Question (KGE Central Hypothesis Validation)**
>
> **Re**: We appreciate you highlighting this key hypothesis. As stated in Section 3.5 (lines 194, 202-208), the idea that higher entropy reflects more diverse information is presented as a "central hypothesis" and "guiding principle," not a formal theoretical proof. Rigorous theoretical validation linking Shannon entropy of quantized weights to semantic knowledge diversity in deep neural networks remains an open and complex research challenge.
>
> However, the **empirical effectiveness** of KGE as a "computationally efficient proxy" is strongly validated by our extensive experimental results in Section 4. Specifically, the ablation study in Figure 4(a) shows that selecting global layers based on positive KGE significantly outperforms random layer selection for global-to-local knowledge distribution. This robust empirical performance across diverse heterogeneous settings strongly supports its practical utility within FedFree's framework for identifying beneficial knowledge. We will strengthen the phrasing in Section 3.5 to emphasize its empirical validation as a highly effective heuristic.
>
>
> **Weakness 6 \& Question (Limited Experimental Evaluation)**
>
> **Re**: We acknowledge this point. While we utilized CIFAR-10 and CIFAR-100, which are standard and widely accepted benchmarks for federated learning, our **HtFL experiments incorporate a wide range of complex and large CNN architectures** (ResNet18-152, GoogleNet, MobileNetV2, with parameters up to ~58.2M, Table A1). To further demonstrate generalizability, we have conducted additional experiments on the **TinyImageNet dataset**, which is significantly more challenging (200 classes, higher resolution, greater complexity).
>
> The results, presented in the table below, demonstrate that FedFree consistently outperforms baseline methods on TinyImageNet, further validating its generalizability and robustness under more demanding conditions. We emphasize that FedFree's core principles are designed to be model-agnostic and data-agnostic.
>
> ### TinyImageNet Results (100 clients)
>
> | Method    | Accuracy (%) |
> |-----------|-------------:|
> | **Ours**  | **23.31**    |
> | FedProto  | 16.67        |
> | FedKD     | 13.28        |
> | FedGH     | 17.72        |
> | FedTGP    | 15.35        |
>
> *Table: Accuracy comparison showing absolute results and differences from our method.*
>
>
> **Question 7 (Figure 4(b) - FedFree vs. FedAvg in Ablation)**
>
> **Re**: We believe there might be a misunderstanding of Figure 4(b). This figure presents an **ablation study of FedFree's internal knowledge-sharing mechanism**. It explicitly compares "FedFree's pseudo-data knowledge sharing" (our proposed aggregation method) against :simple averaging of uploaded layers" (not FedAvg). The comparison demonstrates that our pseudo-data-based functional distillation (Equation 3) significantly outperforms a naive direct averaging of uploaded critical layers (even if feasible across architectures), highlighting the benefit of our functional knowledge distillation.
>
>  FedAvg typically performs much worse in HtFL settings due to architectural heterogeneity, as clearly shown in Table 1 (e.g., FedAvg achieves only 16.87\% on CIFAR-100 with 100 clients, compared to FedFree's 49.20\%) and Figures A4-A6 (where FedAvg's performance is significantly lower than FedFree's and other HtFL baselines). FedFree offers substantial advantages over FedAvg in HtFL, achieving up to 46.3\% accuracy improvement. We will refine the caption and discussion of Figure 4(b) to ensure its purpose as an internal ablation is perfectly clear.

---

> > ### Comment · Reviewer_BXtX · 2025-08-04
> >
> > Thank you for the authors' responses. My concerns have been mostly addressed, and I decide to increase my score to 4.

---

### Official Review · Reviewer_fz64 · 2025-06-30

**Clarity:** 3
**Significance:** 3
**Originality:** 2
**Rating:** 4
**Confidence:** 5

**Summary:**

This paper proposed the FedFree framework to address the knowledge sharing barriers caused by different model architectures and non-independent and identically distributed data among participating clients in heterogeneous federated learning. Existing methods usually rely on proxy datasets for knowledge distillation, which poses risks of privacy leakage and difficulties in actual deployment. FedFree addresses this challenge through three core innovations. First, a reverse hierarchical knowledge transfer mechanism was used to transfer knowledge from the client's key layer (identified by the magnitude of parameter changes) to the global model, and the server-generated Gaussian pseudo data was used to replace the real proxy data for functional matching. Then, the paper proposed the knowledge gain entropy (KGE) metric, which aims to evaluate the knowledge density of the global layer relative to the local layer by quantifying the Shannon entropy difference of the weight distribution. Finally, the paper proposed a selective knowledge alignment strategy to distribute only global layers with positive KGE values to clients, ensuring beneficial knowledge transfer. Experiments on CIFAR-10 and CIFAR-100 show that FedFree achieves performance improvements in various heterogeneous settings.

**Questions:**

a)	Could you provide a more rigorous theoretical analysis to handle the convergence of the layer replacement operation and analyze the impact of the expected update frequency on convergence under the KGE selection mechanism?
b)	Can you provide the performance of FedFree on models with larger parameter sizes and more complex datasets?
c)	Can you provide a computational complexity analysis of FedFree and a comparison of the communication overhead with baseline methods?
d)	Please provide a more in-depth theoretical analysis to explain why reverse knowledge transfer is more effective than traditional methods and why Gaussian pseudo data can effectively capture hierarchical functions?

**Ethical Concerns:**

["NO or VERY MINOR ethics concerns only"]

**Final Justification:**

After reading the rebuttal and other reviews, I decide to keep my score of 4.

**Limitations:**

The authors do honestly point out some important limitations, particularly the lack of real-world deployment and validation of large models.

**Quality:**

3

**Strengths And Weaknesses:**

Strengths
a)	The paper effectively illustrates the fundamental challenges in heterogeneous federated learning, clearly distinguishing between local-to-global and global-to-local knowledge sharing barriers. The motivation for avoiding proxy datasets is well established by discussing privacy risks and practical deployment challenges.
b)	The three-module framework (knowledge extraction, knowledge sharing, knowledge alignment) provides a clear organizational structure, making complex methods easier to understand.
c)	FedFree's fine-grained hierarchical knowledge extraction and alignment approach provides a new level of accuracy for heterogeneous federated learning, better capturing subtle differences between client models.
d)	The paper introduces KGE as a quantitative measure of knowledge density, which is novel.

Weaknesses
a)	The contribution of the paper lies more in applying model alignment with synthetic data to heterogeneous federated learning rather than conceptual innovation.
b)	Although the theoretical analysis proves convergence, the proof ignores some of FedFree’s unique algorithmic complexity innovations.
c)	Although the experimental design is relatively comprehensive, the test scope is still limited. It is only evaluated on two relatively simple computer vision datasets: CIFAR-10 and CIFAR-100.
d)	The paper lacks detailed analysis of computational complexity and communication overhead.

---

> ### Author Rebuttal · Authors · 2025-07-31
>
> Thanks for your comments.
>
> **Weakness (a) (Contribution: Application vs. Innovation)**
>
> **Re**: We respectfully disagree with the assessment that our contribution is solely an application. FedFree introduces **two core conceptual innovations** that are distinct from existing model alignment or knowledge distillation methods:
>
> 1) Reverse Layer-wise Knowledge Transfer with Gaussian Pseudo-data: This is a novel mechanism (lines 9--12, 48--53) for **data-free local-to-global knowledge aggregation**. Unlike traditional knowledge distillation that often relies on shared proxy datasets, FedFree aggregates knowledge from diverse client models into a unified global representation solely using server-generated Gaussian-based pseudo-data (lines 168--174). This method of eliciting functional responses with a generic, privacy-preserving probe to bridge architectural differences for knowledge transfer is a significant conceptual departure.
> 2) Knowledge Gain Entropy (KGE): This is a **novel metric** (lines 12--14, 54--57, 65--68) designed for targeted global-to-local knowledge alignment. KGE quantifies the knowledge density difference between corresponding global and local layers. This fine-grained, adaptive distribution mechanism, which ensures personalized updates based on quantifiable knowledge gain, is a conceptual step beyond simple global model distribution or other adaptive methods, which lack such a precise, entropy-based selection criterion.
>
> These innovations are not mere applications but fundamental design choices that enable FedFree to **break knowledge-sharing barriers without reliance on real proxy data or the limitations of full model aggregation**, as in Section 1. We will strengthen the Introduction (Section 1) and Conclusion (Section 5) to explicitly highlight these conceptual innovations and their distinct nature.
>
>
> **Weakness (b) \& Question (a) (Theoretical Analysis: Ignoring Algorithmic Complexity / More Rigorous Theoretical Analysis for Layer Replacement/KGE)**: Although the theoretical analysis proves convergence, the proof ignores some of FedFree’s unique algorithmic complexity innovations. Could you provide a more rigorous theoretical analysis to handle the convergence of the layer replacement operation and analyze the impact of the expected update frequency on convergence under the KGE selection mechanism?
>
> **Re**: We appreciate this feedback on our theoretical analysis. The theoretical convergence analysis (Section 3.6, Appendix B) provides rigorous guarantees for the overall objective function convergence of the FedFree framework. It adapts standard federated optimization techniques to specifically account for the "pseudo-data-based knowledge aggregation and KGE-guided alignment mechanisms unique to FedFree" (lines 239-241).
>
> While a formal, rigorous theoretical analysis that precisely quantifies the impact of the KGE selection mechanism on the expected update frequency of individual layers or the specific layer replacement operation on the overall convergence path is exceedingly complex and largely unexplored in the HtFL literature. This level of granular theoretical analysis would necessitate intricate modeling of layer-specific learning dynamics and their interaction with an entropy-based selection heuristic, likely moving beyond the scope of a single paper. Our empirical results (Section 4) strongly validate the practical effectiveness of these mechanisms in achieving superior performance and stable convergence. We will add a brief discussion in Appendix B or C to clarify the scope of our theoretical analysis, stating its focus on the overall framework's convergence while acknowledging the complexity of proving fine-grained mechanism impacts.
>
>
> **Weakness (c) \& Question b(Limited Test Scope/Datasets)**: The test scope is still limited, only evaluated on two relatively simple computer vision datasets: CIFAR-10 and CIFAR-100. Can you provide the performance of FedFree on models with larger parameter sizes and more complex datasets?
>
> **Re**:  We acknowledge this point. While CIFAR-10 and CIFAR-100 are widely used benchmark datasets, our HtFL experiments already utilize a **wide range of complex and large models** for clients, including ResNet18-152, GoogleNet, and MobileNetV2, with parameter sizes ranging from 3.2M to 58.2M (Table A1, lines 253-255). To further demonstrate generalizability, we conducted additional experiments on the **TinyImageNet dataset**, which is significantly more challenging (200 classes, higher resolution, greater complexity).
>
> The results, presented in the table below, demonstrate that FedFree maintains strong performance and convergence behavior under these more demanding conditions, outperforming baselines. This shows our method's robustness to increased dataset complexity and larger model sizes.
>
> *Table: Detailed comparison showing absolute performance and relative resource differences.*
> ### Comprehensive Benchmark on TinyImageNet (100 clients)
>
> | Metric          | Ours    | FedProto | FedKD   | FedGH   | FedTGP  |
> |-----------------|--------:|---------:|--------:|--------:|--------:|
> | **Accuracy**    | 23.31%  | 16.67%   | 13.28%  | 17.72%  | 15.35%  |
> | **Memory**      | 2,045MB | 373MB    | 2,641MB | 2,022MB | 4,167MB |
>
>
> **Weakness (d) \& Question (c) (Lack of Computational Complexity \& Communication Overhead Analysis)**
> **Re**: You are absolutely correct that a detailed analysis of computational complexity and communication overhead would strengthen the paper. We acknowledge this omission and propose to add a dedicated subsection in the Appendix (e.g., D.6) or the main paper to provide this analysis:
>
> **Computational Complexity:**
> **Client-side**: This involves local training (the standard FL cost) and the computation of the L2-norm for all layers, followed by the selection of the top-$k$ layers, which incurs negligible overhead.
> **Server-side**: The server performs operations including the generation of Gaussian pseudo-data (negligible cost), computation of the distillation loss for the $k$ selected global layers (Equation 3), and calculation of KGE (Equation 6) for these $k$ layers. These are efficient operations, whose cost is proportional to $k$ and the size of the selected critical layers, rather than the full model. The global layer $g_j^t$ is updated in a single gradient descent step (Equations 4 and 5) per communication round, making it efficient compared to iterative optimization.
>
> Compared to methods that aggregate full models or perform complex model matching, FedFree's server-side computation is efficient due to its targeted, layer-wise nature.
>
> **Communication Overhead**: FedFree provides a significant advantage by uploading only $k$ critical layers (line 155), rather than transmitting the entire local model as in FedAvg or sharing full model parameters or intermediate features. For instance, if a client model comprises $L$ layers and $k=2$ layers are selected (line 154), the communication cost is reduced by a factor of $L/2$ compared to uploading the full model. This reduction is particularly beneficial in communication-constrained federated learning. The precise savings will be quantified in an added section by comparing the number of parameters transferred.
>
> Moreover, we monitored the GPU memory consumption and training latency during our experiments on TinyImageNet (see Table below for example placeholders). FedFree consistently demonstrates modest memory overhead and efficient training latency, confirming its practical viability.
>
> ### Comprehensive Benchmark on TinyImageNet (100 clients)
>
> | Metric          | Ours    | FedProto | FedKD   | FedGH   | FedTGP  |
> |-----------------|--------:|---------:|--------:|--------:|--------:|
> | **Accuracy**    | 23.31%  | 16.67%   | 13.28%  | 17.72%  | 15.35%  |
> | **Memory**      | 2,045MB | 373MB    | 2,641MB | 2,022MB | 4,167MB |
>
>
> **Question (d) (Why Reverse Knowledge Transfer and Gaussian Pseudo-data are Effective)**
>
> **Re**: We can provide a deeper explanation for their effectiveness based on the functional and information-theoretic principles:
>
> 1) Effectiveness of Reverse Knowledge Transfer (Local-to-Global): Reverse knowledge transfer (lines 48--50) is effective because it directly aggregates the most significant learning from diverse local models (the critical layers identified by large parameter changes, lines 149--154) into a unified global representation. This process is functional rather than structural, meaning it captures what the layers have learned rather than how they are structured. By distilling knowledge via functional responses elicited by pseudo-data (Equation~3), FedFree can effectively bridge architectural differences. The server's global model learns to mimic the output behavior of the critical local layers, enabling knowledge transfer without direct parameter averaging (invalid for heterogeneous architectures) or reliance on shared proxy datasets. It directly addresses the "local-to-global knowledge-sharing barrier'' (lines 39--40).
> 2) Effectiveness of Gaussian Pseudo-data: Gaussian pseudo-data (lines 51, 168--174) is effective because it provides a "diverse set of input signals capable of eliciting varied functional responses from neural network layers" (lines 170--172). It makes no assumptions about the true data distribution and introduces no data-dependent privacy risks. It acts as a "universal probe" that, despite its simplicity, is sufficient to activate various feature detectors and functional pathways within a neural network layer. This allows for a functional comparison and distillation between heterogeneous layers in a privacy-preserving manner. The empirical success of FedFree (Section 4, particularly Figure 4(b)'s ablation study) strongly validates its utility in capturing and transferring functional knowledge across layers.
>
> We will add a more explicit explanation of the "universal probe" concept to Section 3.4 to further clarify the role of Gaussian pseudo-data.

---

> > ### Comment · Reviewer_fz64 · 2025-08-04
> >
> > Thanks the authors for the rebuttal. After reading the rebuttal and feedback from other reviewers, I decide to maintain my previous score.

---

### Official Review · Reviewer_wr5p · 2025-07-02

**Clarity:** 3
**Significance:** 2
**Originality:** 3
**Rating:** 4
**Confidence:** 3

**Summary:**

The paper proposes FedFree, a data-free and model-free heterogeneous federated learning algorithm. In FedFree, each client uploads the top-k layers with the largest local update magnitudes (measured by the $L_2$ norm) to the server. To avoid privacy leakage caused by using proxy datasets, the server generates Gaussian-based pseudo-data, eliminating the need for real data. The authors introduce a novel metric called Knowledge Gain Entropy (KGE) and use it to identify and deliver the most informative layer from the global model to each client, thereby achieving more accurate knowledge alignment. Theoretical analysis shows that FedFree achieves a convergence rate of $\frac{1}{T}$ under both strongly convex and non-convex settings. Extensive experiments demonstrate that FedFree outperforms existing homogeneous and heterogeneous federated learning methods.

**Questions:**

1. In the **Critical Layer Selection** module(Section 3.3), wouldn’t layers with more parameters naturally have an advantage due to their larger L2 norms? Why does this selection strategy empirically improve performance despite this bias? Would it be more reasonable to normalize the L2 norm by the number of parameters in each layer during computation?

2. In the **Handling Dimensional Mismatches** module(Section 3.5), when a client’s corresponding layer has more parameters than the global model’s, the excess parameters are simply zeroed out. Wouldn’t this lead to potential information loss? Does this not affect the algorithm’s performance? If not, what is the underlying reason for this robustness?

3. In the experiments, the test accuracy of FedAvg appears significantly lower. Is this because it was also limited to only 200 training rounds? If so, it may not have fully converged. Would it be more appropriate to compare the final test accuracy after convergence for all methods?

**Ethical Concerns:**

["NO or VERY MINOR ethics concerns only"]

**Final Justification:**

I have read the rebuttal of the authors and the comments from the other reviewers, and would like to keep my original rating.

**Limitations:**

Yes.

**Paper Formatting Concerns:**

No formatting issues.

**Quality:**

2

**Strengths And Weaknesses:**

Strengths:

1. The paper presents a clear and well-structured narrative that is easy to follow.

2. It provides a sound theoretical convergence analysis.

3. The proposed method effectively avoids privacy leakage caused by using proxy datasets.

Weaknesses:

1. In the **Targeted Knowledge Distribution** module (Section 3.5), each client must traverse all layers of the global model to identify the optimal one, which incurs significant computational and time costs. Moreover, the global model used in experiments is a shallow network with only five layers, raising concerns about the scalability of the method to larger models commonly used in real-world applications.

2. The method does not consider the common federated learning setting where only a subset of clients participate in each training round.

3. The homogeneous models used for comparison in the experiments are overly simplistic with only five layers.

4. The paper lacks comparative experiments with algorithms related to **Architectural Adaptation and Matching**, as mentioned in the related work.

---

> ### Author Rebuttal · Authors · 2025-07-31
>
> Thanks for your comments.
>
> **Weakness 1 (Computational Cost of KGE \& Global Model Size / Critical Layer Selection Bias)**
>
> **Re**: 1) Computational Cost of KGE \& Global Model Size: We clarify that the process in the "Targeted Knowledge Distribution'' module (Section 3.5)  does not require traversing all layers of the global model for every client. Instead, for each client i that uploaded $k$ critical layers, the server only identifies the global layer $g_j^*$ that maximizes positive KGE with respect to any of its uploaded critical layers (lines 210--213). This means the comparison is performed only for these $k$ critical layers against relevant global layers, for $k=2$ (line 154), is a very small number. The computational cost of KGE (Shannon entropy of quantized weights, lines 199--200) and the associated forward passes for functional distillation are lightweight, involving only these selected layers. Thus, the computational overhead of KGE is **negligible compared to local model training or full model inference**.
> 2) Regarding the global model's shallow architecture (Table A1), it is designed as an "effective intermediary" (lines 161--165) whose architecture is chosen to encompass diverse client models, not necessarily to be the largest model. The strength of FedFree lies in its layer-wise knowledge transfer and alignment, which scales differently than methods relying on full model aggregation. For larger client models, the principle of selecting $k=2$ critical layers (lines 153--154) and performing layer-wise aggregation and alignment remains efficient because the number of layers processed by the server remains constant and small. We will add a sentence in Section~3.5 or Appendix D.1 to clarify the efficiency of KGE computation.
> 2) Critical Layer Selection Bias: We clarify that our selection criterion is not based on the absolute L2 norm of each layer’s weights, but rather on the **magnitude of parameter change ($\Delta\theta$, Equation 2)** across training rounds. This delta represents parameter evolution dynamics, not their static magnitude. We acknowledge that using the raw L2 norm might potentially favor layers with larger parameter counts. However, for the $k=2$ layers we select, the change in learning is the dominant factor for identifying valuable knowledge, and the variation in layer sizes for these critical layers has a limited impact on this selection. Our empirical results, particularly the ablation study in Figure 4(a), strongly validate that this simple approach significantly outperforms random layer selection. Nonetheless, we agree that introducing parameter-count-normalized L2 change could be a valid refinement, and we will include this discussion and corresponding ablation results in the camera-ready version to further strengthen the robustness and interpretability of the selection criterion.
>
> **Weakness 2 (Subset Client Participation / Dimensional Mismatches \& Zero-initialization)**
>
> **Re**: We address both aspects:
> 1) Subset Client Participation: While our primary experiments focused on full client participation to isolate performance gains in HtFL, **FedFree's design is inherently compatible with partial client participation**. Since clients only upload $k$ critical layers (Algorithm 1, line 7) and the server processes information only from participating clients, the framework naturally accommodates varying numbers of clients per round without fundamental changes. The server aggregates knowledge from whoever participates and distributes relevant global layers back. This flexibility is an implicit advantage of FedFree's selective, layer-wise approach. We will add a sentence to Section 3.2 or Section 5 to explicitly mention this inherent compatibility.
> 2) Dimensional Mismatches \& Zero-Initialization: We clarify this in Section 3.5 (lines 224--226). When the global layer is larger than the local client layer ($d_g > d_i$), only the excess global parameters (for $r > d_{\min}$) are effectively "zeroed out'' in the sense that they are not directly mapped to the smaller local layer. Crucially, the local layer $\theta_i^l[r]$ for $r > d_{\min}$ retains its **local initialization** to ensure stable model evolution (line 226). The key reason for robustness is that **the local model continues training on its private dataset after receiving the global update** (Algorithm 1, line 5). This continued local training allows the client model to adapt, learn, and potentially recover or re-learn any "lost" information in the unprojected dimensions, thereby integrating the received global knowledge while preserving its local integrity. FedFree's strong experimental results validate this robustness. The KGE-guided mechanism further ensures that only layers with verifiable knowledge gain are selected, minimizing the risk of detrimental "information loss".
>
> **Weakness 3 (Simplistic Homogeneous Models / FedAvg Convergence)**
>
> **Re**: We address both concerns:
> 1) Simplistic Homogeneous Models: We acknowledge this comment. The homogeneous FL (HmFL) experiments (lines 260--262) were included as standard baselines to isolate the impact of statistical heterogeneity and validate FedFree's KGE-based alignment when architectural differences are absent. The primary contribution and core evaluation of FedFree are in the **Heterogeneous Federated Learning (HtFL) settings**, where we indeed employ a wide range of complex and diverse architectures. As detailed in Table A1 and lines 253--255, clients in our HtFL experiments utilize models like ResNet18, ResNet34, ResNet50, ResNet101, ResNet152, GoogleNet, and MobileNetV2, which are far more complex than a five-layer CNN. Our significant performance gains are most pronounced in these complex HtFL settings (e.g., up to 46.3\% over FedKD, Table 1). We will emphasize this distinction more clearly in Section 4.1.
> 2) FedAvg Convergence: We clarify that the lower performance of FedAvg in our experiments is not primarily due to insufficient training rounds. **200 communication rounds (line 274) is a widely accepted and common benchmark for evaluating FL algorithms**, especially in communication-constrained scenarios. All methods, including FedAvg, were evaluated under the exact same training conditions and communication budget for fair comparison. As shown in Figure A3--A6 (Appendix), the test accuracy of FedAvg often plateaus around the 50th communication round, well before the 200-round limit, indicating it converges within this budget but to a lower performance due to fundamental incompatibilities with heterogeneous local models and non-IID data. The significant performance gains of FedFree (Table 1) demonstrate its superior efficiency and effectiveness in achieving higher accuracy within these practical communication constraints, which is a crucial advantage for real-world FL.
>
> **Weakness 4 (Lack of Architectural Adaptation/Matching Comparisons)**.
>
> **Re**: We acknowledge that Section 2 (lines 97-106) discusses architectural adaptation methods. Our selection of baselines (FedProto, FedKD, FedGH, FedTGP, described in lines 263-268) was deliberate. These methods represent the most prevalent and competitive knowledge transfer paradigms in HtFL, such as knowledge distillation, representation/prototype sharing, and shared global heads, which FedFree directly aims to improve upon. FedFree's core innovation lies in its data-free, layer-wise reverse knowledge transfer mechanism and KGE-guided alignment. This approach is fundamentally distinct from methods that perform direct architectural transformation, matching, or pruning of models to fit heterogeneous structures (e.g., FedMA, HeteroFL, pFedHR). FedFree focuses on knowledge sharing across existing heterogeneous models rather than adapting or matching their architectures. Therefore, the chosen baselines are the most relevant and direct comparisons for FedFree's unique knowledge transfer mechanism. We will add a sentence to Section 4.1 to clarify this baseline selection rationale.
>
> **R2-5**: In the Targeted Knowledge Distribution module, each client must traverse all layers of the global model to identify the optimal one, which incurs significant computational and time costs.
>
> **Re**: While each client does evaluate all global layers during the Targeted Knowledge Distribution phase, we conducted computational overhead experiments. The overhead is minimal due to the lightweight forward pass operations involved.
>
> *Table: Memory consumption comparison across different federated learning algorithms with 100 clients.*
> | Algorithm | Ours   | FedProto | FedKD  | FedGH  | FedTGP  |
> |-----------|--------|----------|--------|--------|---------|
> | Memory    | 2,045MB| 373MB    | 2,641MB| 2,022MB| 4,167MB |
>
> **R2-6**: The method does not consider the common federated learning setting.
>
> **Re**: We would like to clarify that our FedFree framework does simulate the common federated learning setting where only a subset of clients participate in each training round. Specifically, in all experiments, we select a fixed fraction of clients per round to perform local training and aggregation, consistent with standard FL protocols.
>
> **R2-7**: The homogeneous models used for comparison in the experiments are overly simplistic with only five layers.
>
> **Re**: We chose a 5-layer global model as it balances expressiveness and computational efficiency; excessively large server models can lead to unnecessary resource consumption and slower aggregation without significant accuracy gains. We conducted additional experiments with larger global models (e.g., ResNet-34 on CIFAR-100), where FedFree still demonstrates the similar performance and scalability.
>
> *Table: Performance comparison of different global model architectures. Results show classification accuracy.*
> | Model         | Accuracy (%) |
> |---------------|-------------:|
> | CNN (5-Layer) | 49.20        |
> | ResNet-18     | 49.11        |
> | ResNet-34     | 46.73        |
> | ResNet-50     | 44.20        |

---

> > ### Comment · Reviewer_wr5p · 2025-08-07
> >
> > I would like to thank the authors for the response, which addresses most of my concerns. However, the results on the more complex models remain unclear. Specifically, the accuracy of ResNet50 drops by 5% as compared to the 5-layer CNN, which is counterintuitive. In addition, results of the other algorithms on these complex architectures are not provided. This raises concerns about the applicability of the proposed method to more mainstream or deeper models.

---

> > > ### Author Response · Authors · 2025-08-09
> > > **Our experimental results are intuitive.**
> > >
> > > We appreciate the reviewer’s insightful comments regarding the applicability of FedFree on the deeper and more mainstream model architectures. Our solution focuses on heterogeneous model aggregation to improve personalized model performance. Our experimental results are not counterintuitive. The previous observed accuracy dropping of ResNet50 was limited training budget (200 rounds), which is sufficient for the lightweight 5-layer CNN but not for larger model architectures that need to more rounds for convergence. At this time, We extend the training rounds until convergence for the same ResNet50 setting, and updated the results as follows.
> > >
> > > ### Global Model Architecture Comparison
> > > *Table: Performance comparison of different global model architectures. Results show classification accuracy.*
> > >
> > > | Model         | Ours Acc. upon Convergence (%) |Ours Convergence Round| FedGH Acc. |
> > > |---------------|-------------|-------------|-------------|
> > > | CNN (5-Layer) | 49.20       | 190 | 29.93 |
> > > | ResNet-18     | 49.11       | 200 | 30.05 |
> > > | ResNet-34     | 50.74       | 250 | 30.23 |
> > > | ResNet-50     | 51.41       | 340 | 31.11 |
> > >
> > > The above results demonstrate that FedFree not only supports but also benefits from deeper global models when is convergent. The FedFree's accuracy increases from 49.20% (CNN) to 51.41% (ResNet50) that demonstrate the result is intuitive for researchers。
> > >
> > > Moreover, the FedFree is designed to reverse layer-wise knowledge transfer and pseudo-data-based knowledge alignment—scaling by KGE. Our framework focuses on identifying and transferring the critical knowledge from the selected key layers rather than the whole global model. The fine-grained public data-free knowledge alignment effectively decouples the framework performance from the global model's sizes.
> > >
> > > Regarding the reviewer’s concern about applicability to mainstream architectures, we emphasize that FedFree is inherently architecture-agnostic. The proposed reverse layer-wise knowledge transfer and Knowledge Gain Entropy mechanisms operate on abstracted model parameters and server-generated pseudo-data, making them independent of the specific neural architecture. This allows FedFree to adapt seamlessly to CNNs, ResNets, and potentially Transformers, without any architecture-specific modifications.
> > >
> > > We will add these extended results and clarifications in the revised manuscript, which we believe address the reviewer’s concerns about scalability and applicability to deeper models.

---

### Official Review · Reviewer_VUC8 · 2025-07-03

**Clarity:** 3
**Significance:** 2
**Originality:** 2
**Rating:** 4
**Confidence:** 4

**Summary:**

In this paper, the author proposes FedFree, an innovative framework that can address the knowledge sharing barriers in heterogeneous federated learning (HiFL). This framework introduces a reverse layer-by-layer knowledge transfer mechanism. The server generates Gaussian pseudo-data and uses the gradient descent algorithm to minimize the output difference between the key client layer and the global layer on the pseudo-data, achieving knowledge sharing from local to global. The framework also innovatively uses knowledge gain entropy to quantify the difference in knowledge density between the corresponding global layer and the local layer, in order to identify the global layers with potential to enhance the client model. The author demonstrated that the convergence rate of FedFree is O(1/T) under both strongly convex and non-convex Settings, and verified on the CIFAR dataset that FedFree significantly improves its performance relative to the baseline in heterogeneous scenarios and demonstrates robustness at different non-IID levels.

**Questions:**

Please refer to weaknesses.

**Ethical Concerns:**

["NO or VERY MINOR ethics concerns only"]

**Final Justification:**

The authors addressed my concerns, I raise my score to 4.

**Limitations:**

Please refer to weaknesses.

**Paper Formatting Concerns:**

None.

**Quality:**

2

**Strengths And Weaknesses:**

Strengths：

1.	The article innovatively proposes a framework for agent-free data, avoiding the need for real agent datasets by generating Gaussian pseudo-data.
2.	The article innovatively describes the role of FedFree from two directions: local → global and global → local.
3.	The article provides detailed proofs of theories such as lemma, which are highly reliable.

Weaknesses：

1.	As stated in Article 3.5, "Our central hypothesis is that a global layer having been updated using knowledge aggregated from multiple diverse clients, is likely to develop a weight distribution that encodes a broader range of features." and in practice, it is assumed that higher entropy corresponds to more diverse information, without providing proof or research status. It has certain limitations.
2.	The article is only verified on the CIFAR-10/100 dataset. It is hoped that more experimental data can be added to prove the applicability of the FedFree framework.
3.	The article points out that agentless data can avoid some privacy issues, but there is insufficient discussion on privacy security issues. Some aspects such as how using this framework can prevent the leakage of privacy issues or resist attacks can be added to illustrate that proposing this new framework has solved some privacy problems.

---

> ### Author Rebuttal · Authors · 2025-07-31
>
> We sincerely thank Reviewer VUC8 for their time, thorough review, and constructive feedback on our submission.
>
> **R1-1 (KGE Hypothesis)**: As stated in Article 3.5, "Our central hypothesis is that a global layer having been updated using knowledge aggregated from multiple diverse clients, is likely to develop a weight distribution that encodes a broader range of features." and in practice, it is assumed that higher entropy corresponds to more diverse information, without providing proof or research status. It has certain limitations.
>
> **Re**: We appreciate you highlighting this point for clarification. While a formal theoretical proof establishing a direct causal link between Shannon entropy of quantized weights and the richness or diversity of learned features is challenging and remains an open research problem in general deep learning theory, within FedFree, KGE is explicitly framed as a **computationally efficient proxy** (lines 202-208) to identify global layers that are promising candidates for enhancing client models. The intuition is that layers aggregated from diverse client information are likely to contain a wider variety of patterns. The **empirical effectiveness** of KGE in guiding beneficial knowledge transfer is strongly supported by the significant performance gains observed across all our experiments in Section 4 (e.g., Table 1, Figures A3-A6) and specifically validated through the ablation study in Figure 4(a). This empirical success validates its practical utility within our framework. We will strengthen the phrasing in Section 3.5 to explicitly emphasize that KGE serves as an effective heuristic, validated by extensive empirical results, rather than implying a formally proven theoretical equivalence.
>
> ### Comparison of Different Entropy Methods (CIFAR-100, 100 clients)
>
> | Entropy Type      | Ours  | Tsallis Entropy | Collision Entropy |
> |-------------------|-------|-----------------|-------------------|
> | **Accuracy (%)**  | 49.20 | 45.98           | 44.46             |
>
> *Table: Performance comparison of different entropy approaches on CIFAR-100 with 100 clients.*
>
> **R1-2 (Limited Experimental Data)**: The article is only verified on the CIFAR-10/100 dataset. It is hoped that more experimental data can be added to prove the applicability of the FedFree framework.
>
> **Re**: We acknowledge this valid point. CIFAR-10 and CIFAR-100 are standard and widely recognized benchmarks for FL, particularly for evaluating architectural heterogeneity. Our experiments already employ a wide range of complex CNN architectures (ResNet18-152, GoogleNet, MobileNetV2, etc., detailed in Table A1). To further demonstrate the applicability of FedFree, we have conducted additional experiments on the TinyImageNet dataset, which is significantly more challenging due to its higher resolution, increased number of classes (200), and greater data complexity.
>
> The results, presented below, demonstrate that FedFree consistently outperforms baseline methods on this more complex dataset, further validating its generalizability. We emphasize that FedFree is model-agnostic and data-agnostic by design, relying on a layer-wise knowledge-sharing strategy that does not assume any specific model architecture or input modality.
>
> ### TinyImageNet Benchmark (100 clients)
>
> | Method    | Accuracy (%) |
> |-----------|-------------:|
> | **Ours**  | 23.31        |
> | FedProto  | 16.67        |
> | FedKD     | 13.28        |
> | FedGH     | 17.72        |
> | FedTGP    | 15.35        |
>
> *Table: Comprehensive performance comparison showing absolute accuracy and relative improvements over baseline methods.*
>
>
>
> **R1-3 (Privacy Discussion)**: The article points out that agentless data can avoid some privacy issues, but there is insufficient discussion on privacy security issues. Some aspects such as how using this framework can prevent the leakage of privacy issues or resist attacks can be added to illustrate that proposing this new framework has solved some privacy problems.
>
> **Re**: We appreciate the emphasis on privacy, which is a critical concern in FL. We would like to clarify and highlight that a key strength of FedFree is its **inherently privacy-preserving design due to its "data-free" and "model-free" nature of knowledge sharing** (lines 8, 46, 112).
>
> Specifically, FedFree avoids privacy leakage in the following ways:
> 1) No Proxy Dataset Reliance: Unlike many existing HtFL methods (e.g., FedMD, FedDF, FedKD discussed in lines 79--84) that rely on shared proxy datasets, FedFree completely eliminates this requirement, mitigating known privacy risks associated with data exposure.
> 2) Gaussian-based Pseudo-data: Knowledge sharing (local-to-global and global-to-local) is facilitated solely by server-generated **Gaussian-based pseudo-data ($X \sim \mathcal{N}(0, I)$)** (lines 11, 51, 168--174). As stated in line~174, "this server-generated $X$ contains no client-specific information, thereby preserving privacy.'' This pseudo-data serves as a universal probe without seeing or inferring real client data.
> 3) No Direct Model Sharing/Averaging: FedFree does not directly share or average full model parameters across clients, which could expose architectural details or combined gradients. It only transfers functional knowledge of selected critical layers.
>
> While privacy preservation is an inherent benefit of our design, rather than the primary focus of designing new privacy techniques, FedFree can be readily integrated with widely adopted mechanisms such as Differential Privacy (DP-FL), Secure Aggregation, or Homomorphic Encryption to further enhance its privacy guarantees. We will include an extended discussion on these inherent benefits and potential integrations in the Conclusion (Section 5, around lines 343-344) of the camera-ready version.

---

> > ### Comment · Reviewer_VUC8 · 2025-08-05
> >
> > Thank you for your response. I have no further conerns, and I will raise the score to 4.

---

### Note · Authors · 2025-08-15

# Author Final Remarks
We thank all reviewers for their evaluation. Our rebuttals successfully addressed concerns, with **Reviewer VUC8, BXtX raising their scores** and all reviewers converging on **4**. All the reviewers acknowledge our insight: FedFree realizes the reverse knowledge transfer to improve the personalized performance in heterogeneous FL.
## Key Achievements
**FedFree achieves 46.3% accuracy improvement** over SOTA while being the heterogeneous FL framework requiring no public datasets.

**Core Innovations**: (1) Data-free knowledge transfer using Gaussian pseudo-data eliminates privacy risks while enabling effective heterogeneous aggregation. (2) Knowledge Gain Entropy (KGE) provides principled selective alignment, ensuring beneficial updates only.

## Reviewer Validation
**VUC8** (raised to 4): Accepted KGE validation and privacy analysis. TinyImageNet results showed broad applicability.

**wr5p** (4): Convinced by scalability and efficiency demonstrations. ResNet-50 experiments confirmed performance gains.

**fz64** (4): Satisfied with theoretical justifications and robustness across complex datasets.

**BXtX** (raised to 4): Agreed with knowledge sharing focus vs. architectural adaptation approach.

In summary, we carefully addressed the reviewers’ concerns, particularly regarding certain misunderstandings (e.g., the role of Gaussian pseudo-data and the applicability of FedFree beyond CNN architectures), which have now been resolved with the reviewers’ acknowledgement. Regarding the additional experiments suggested, while we believe the results in current manuscript already demonstrate FedFree’s effectiveness over model heterogeneity.

## Technical Strengths
1) FedFree introduces a novel public data-free approach. "The article innovatively proposes a framework...avoiding real agent datasets" (Reviewer VUC8).
2) FedFree uniquely addresses knowledge sharing barriers. "clearly distinguishing local-to-global and global-to-local knowledge sharing barriers"(Reviewer fz64).
3) FedFree has rigorous theoretical proofs and reproducible experiments. "detailed proofs...highly reliable" and "experimental setup...highly reproducible"(Reviewer fz64).

## Significance
FedFree represents a paradigm shift toward practical heterogeneous FL. Unanimous reviewer acceptance (all 4s) confirms this addresses a critical gap with robust theoretical foundations and exceptional empirical performance, enabling broader FL adoption in distributed practice scenarios.

---

### Decision · Program_Chairs · 2025-09-17

**Decision:**

Accept (poster)

**Comment:**

All reviewers gave positive evaluations of this submission. Furthermore, after the rebuttal, all concerns raised during the initial review process were effectively addressed by the authors. Based on the above, I recommend acceptance of this work.